# Manipulating and visualizing the dynamic aggregation-induced emission within a confined quartz nanopore

Yi-Lun Ying[1], Yuan-Jie Li[1], Ju Mei[1], Rui Gao [1], Yong-Xu Hu[1], Yi-Tao Long[1] & He Tian [1]

Aggregation-induced emission (AIE) as a unique photophysical process has been intensively explored for their features in fields from optical sensing, bioimaging to optoelectronic devices. However, all AIE luminogens (AIEgens) hardly recover into the initial dispersed state after illuminating at the ultimate aggregated state, which limits AIEgens to achieve reversible sensing and reproducible devices. To real-time manipulate the emission of AIEgen, here we take the advantage of confined space in the quartz nanopore to achieve a nanopore-size-dependent restriction of AIEgens for reversible conversions of "on-to-off" and "off-to-on" emission. By electrochemically manipulating 26 fL AIEgen solution inside nanopore confinement, AIE illuminates while moves along nanopore from the constricted tip to inside cavity at a velocity of 1.4–2.2 µm s$^{-1}$, and vice versa. We further apply this dynamic manipulation for a target delivery of AIEgen into single cells, which opens up new possibility to design powerful and practical AIE applications.

---

[1] Key Laboratory for Advanced Materials & School of Chemistry and Molecular Engineering, East China University of Science and Technology, 200237 Shanghai, P. R. China. Correspondence and requests for materials should be addressed to Y.-L.Y. (email: yilunying@ecust.edu.cn)

The emerged aggregation-induced emission luminogens (AIEgens) show the characteristics of high emission efficiency in the aggregated state with advantages of high photostability and large Stokes' shift[1,2]. The most recognized mechanism of AIE has been rationalized to be the restriction of intramolecular motions. Take the 1,1-dimethyl-2,3,4,5-tetra-phenylsilole (DMTPS) derivate as an example (Fig. 1a), the proposed mechanism indicates that the motions of phenyl peripheries in good solvent such as acetonitrile would efficiently consume the excited-energy and result in no or weak emission, while the intramolecular motions are effectively restricted in the poor solvent such as aqueous solution leading to the fluorescence emission through the radiative decay of excited-state energy. Therefore, turning the AIEgens into emissive state could be used to achieve a wide range of favorable applications, such as bio/chemosensing[3,4], bioimaging[5,6], and optoelectronic devices[7–9]. The strategies of turn-on principle include (i) self-assembly with the target to form aggregates[10], (ii) restrict intermolecular motion by the interaction with specific chemical species[11], and (iii) cleavage of dissolution-promoting ligands to induce the aggregates formation of the AIE residue in the poor solvent[12,13]. All these principles are based on measuring two optical states of AIEgens which are the initial dispersed state and the ultimate aggregated state. It is still hard for AIEgens to recover into the initial dispersed state once generating the ultimate aggregated state. This drawback limits AIEgens to achieve reversible sensing and reproducible optical device. Although AIEgens has been confined into the porous organic frameworks[14], two-dimensional (2D) nanomaterials[15], confined chiral nanotubes[16], and nano-channels of metal-organic frameworks[17], these physical restrictions only induce the emission of AIEgens rather than the reversible manipulation of the emission in "on-to-off" and "off-to-on" manner. Moreover, there is no approach and direct evidences for tracing the dynamic emission of AIEgen since it is challenging to manipulate the movement of AIEgen from good solvent to poor solvent in real time.

The nanopore offers the unique nanoscale environment partitioned from the surrounding bulk space, which features the high sensitivity and high spatial resolution[18–21]. The lumen of nanopore provides a confined space for taking places the chemical reactions, intermolecular interactions as well as the fluorescence emission from labeled biomolecules or $Ca^{2+}$-activated dyes to take place[22–25]. More importantly, the nanopore technology achieves electrochemically manipulation of the molecules or solutions inside confinement[26,27]. Therefore, confining the AIEgens into the nanopore enables the numbers of AIE molecules under study to be electrochemically tuned and controlled for aggregation in ways not possible with bulk system.

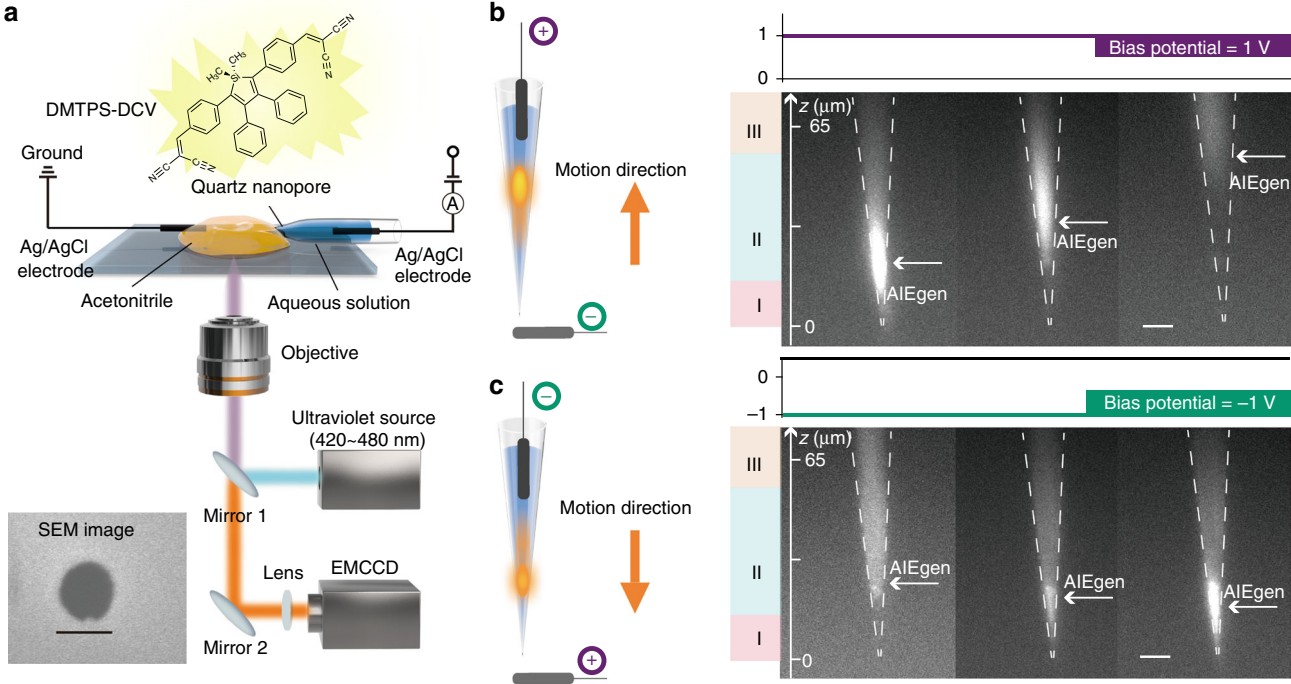

**Fig. 1** Visualizing the dynamic emission of DMTPS-DCV by electrochemically confinement. **a** The instrumentation for the visualization of DMTPS-DCV emission inside a quartz nanopore. The quartz nanopore is placed on the designed cover glass (See Supplementary Figure 1). The inside and outside nanopore filled with aqueous solution and 10 μM DMTPS-DCV acetonitrile solution, respectively. The 10 mM TBAPF6 and 10 mM KCl were used as organic and aqueous electrolyte to conduct the ionic flow. Two Ag/AgCl electrodes were used to apply the potential while the acetonitrile side is defined as virtual ground. The ionic current is recoded by an ultrasensitive electrochemical amplifier (See Supplementary Figure 2). The light of $\lambda = 420$–480 nm was used to excite the DMTPS-DCV. All the fluorescent images were recorded by EMCCD at exposure time of 50 ms and gain of 300. Detailed Schematic configuration for synchronized collection of electrical and optical signals are shown in Supplementary Figure 2. Insert: the SEM image of the quartz nanopore with a diameter of approximately 100 nm. Scale bars, 100 nm. **b** The electrochemical manipulation of DMTPS-DCV at applied bias potential of 1 V. Left: the DMTPS-DCV solution moves from outside nanopore to the inside nanopore at 1 V, leading to an intensive emission. Right: the fluorescent images of the "on-to-off" motion of DMTPS-DCV inside a confined nanopore, which are recorded within 5 s. Scale bars, 10 μm. **c** The electrochemical manipulation of DMTPS-DCV at applied bias potential of −1 V. Left: the applied potential of −1 V drove the movement of DMTPS-DCV aggregates from the wider lumen to the confined tip. Right: Fluorescent images recorded at −1 V, illustrating the "off-to-on" motion of DMTPS-DCV. The bright pixel moves from upper to tip within 35 s. Scale bars, 10 μm. The length of the quartz nanopore shown in each fluorescent image is 74 μm which could be dived into three confinements of I ($z = 0$–16 μm), II ($z = 16$–56 μm) and III ($z > 56$ μm), respectively. The white arrow indicates the strongest location of DMTPS-DCV emission

Herein, we report the visualization of the dynamic emission from an AIEgen by electrochemically manipulating its motion within a confined quartz nanopore, which shows high reversibility and repeatability. Moreover, we apply this dynamic manipulation method for electrochemically delivering the impermeable AIEgen into a single cell.

## Results

**Manipulating the AIE emission within a quartz nanopore**. Here, the quartz nanopore was fabricated based on our previous studies[19,28], which is slightly conical with an outer half angle of 5° (Supplementary Figure 3). The I–V and SEM characterization show the quartz nanopore owns the diameter of approximately to 100 nm (Fig. 1a and Supplementary Figure 4). Then, it is placed in the PDMS cell covered with a quartz slide (Fig. 1a). The DMTPS derivate, 1,1-dimethyl-2,5-bis(4-benzylidenemalononitrile)-3,4-diphenylsilole (DMTPS-DCV), is used as a model AIEgen in our experiments. The synthesis and characterization of DMTPS-DCV were discussed in our previous studies[29,30]. First, DMTPS-DCV was dissolved in acetonitrile whereas it barely exhibits as a weak emitter in the good solvent. Then, the acetonitrile solution of DMTPS-DCV was added into the outside of nanopore whereas the inside was filled with aqueous solution. Note that 10 mM tetra-n-butylammonium hexafluorophosphate (TBAPF6) and 10 mM KCl were used here as electrolyte in the outside and inside solution to conduct the ionic current through the nanopore, respectively. The finite element method simulation (FEM)[19,31] was used to demonstrate that the largest fraction of the bias potential drops at the narrowest tip of nanopore with approximately region of 15 μm in length (Supplementary Figure 5-6), which induced a strong electric filed at the tip confinement for generating the electrochemical surface tension effects. The detailed description of the FEM simulation is shown in Supplementary Note 1. Since the potential drop between two liquid phases could be controlled by a bias potential, one Ag/AgCl electrode was inserted in the quartz nanopore while another Ag/AgCl electrode regarding as virtual ground immersed in the outer solution. Previous studies suggest that the application of voltage across the liquid/liquid interface changes the surface tension, leading to dispensing atto-to-femtolitre of solution in/out the nanopore[27,32]. In our control experiments, the outside acetonitrile solution ingress into the aqueous filled quartz nanopore at a bias potential of 1 V (Supplementary Note 2 and Supplementary Figure 7). Since the AIEgen flow is largely determined by the surface tension effects in the confinement, the extremely high or low KCl concentration prevents the electrochemical control of the fluid motion (Supplementary Figure 8-9 and Supplementary Note 3). Here, the optimized the KCl concentration was set as 10 mM. Then, we applied bias potential to trigger the motion of DMTPS-DCV solution in/out of the nanopore. As setting the optimal bias potential to 1 V (Supplementary Figure 10), the acetonitrile solution of DMTPS-DCV is driven into the inside aqueous solution from the exterior. Under UV irradiation, this action induces the emission of DMTPS-DCV at the tip area of nanopore (Fig. 1b), which could be recorded in real time by the EMCCD camera. The very tip of nanopore (position I in Fig. 1b) was filled with 26 fL acetonitrile as calculated from the control experiment in Supplementary Figure 7. Therefore, the dissolved DMTPS-DCV did not exhibit obvious emission at position I. However, there is a strong emission occurring at the position II of quartz nanopore (Figs. 1b, 2a). This result demonstrates that the nanopore provides the confined space for accumulating the DMTPS-DCV molecules at the internal aqueous solution, which modulates the conversion of DMTPS-DCV to a strong emitter. Since the quartz nanopore can suck 26 fL of acetonitrile, we

assume that 26 fL of 10 μM DMTPS-DCV was driven into the tip of nanopore. Therefore, the number of DMTPS-DCV molecules in the tip of nanopore could be roughly estimated to be ~161,000. As the time elapses, the positive dielectrophoretic force drives the neutral DMTPS-DCV aggregates migrating more deeply into the internal cavity. Consequently, the brightest pixel in the individual frame of fluorescence image shifts to the upper end of the nanopore (Position II in Fig. 1b), which is due to the dynamic aggregation-induced emission process. Since the cylindrically shaped cavity of the nanopore owns a larger volume than its conical tip, the DMTPS-DCV aggregates become more dispersed to exhibit weaker emission at Position III. This dynamic emission at 1 V is defined as the "on-to-off" process.

As stepping the bias potential to −1 V, the DMTPS-DCV solution immediately flowed outside nanopore (Fig. 1c). This flow brings the DMTPS-DCV back to the confined conical tip of nanopore (position II in Fig. 1c), which leads to the higher emission. Consequently, the nanopore was successively lighted from the upper cavity to the bottom tip, which are assigned to the dynamic "off-to-on" emissions. These results further demonstrate that the confinement of nanopore promote the emission of DMTPS-DCV while the wide cavity suppressed its fluorescence behaviors. To confirm our findings, as a control, both the outside and inside of nanopore were filled with the aqueous solution or acetonitrile solution, while the DMTPS-DCV was added into the exterior of nanopore. Both these two experiments show that no bright spot was observed in the fluorescence image by applying either +1 V or −1 V (Supplementary Figure 11-12). Therefore, by taking advantage of asymmetrical nanopore, the intramolecular motions of DMTPS-DCV are dynamically restricted in acetonitrile along with the nanopore constriction. Note that the electric response was synchronized recorded with the optical signals in all the experiments (Supplementary Figure 2). As shown in the Supplementary Movie 1, the ingress of the outside DMTPS-DCV at 1 V leads to an increase of the interface resistance, which exhibits a decrease of the ionic current value. Inversely, the out flow of the DMTPS-DCV solution from the inside nanopore to the outside nanopore at −1 V generates a decreasing of the interface resistance, which suddenly increases the value of ionic current before gradually reaching to a constant value. Therefore, our method achieves real-time manipulation and visualization of the dynamic emission of AIEgens.

**Reversible controlling the AIE emission within a nanopore**. Since the movement of DMTPS-DCV solution could be electrochemically controlled, we repeatedly performed the dynamic "on-to-off" and "off-to-on" emission process within the same quartz nanopore (Fig. 2a and Supplementary Movie 2). The motion of the DMTPS-DCV aggregates was manipulated every 15 s via alternatively applied the bias potential of −1 V or 1 V. Every stepping of 1 V forces the occurrence of the strong emission at position II in tip area while −1 V drives the bright aggregates away from the nanopore tip. Also, the moving distance of the DMTPS-DCV emission is nearly the same during each manipulation, illustrating the high efficiency of our methods. Figure 2b shows the normalized fluorescence intensity at position of $z = 50$ μm from three individual quartz nanopores, which illustrating the high stability and high repeatability of our strategy. By tracing the motion of brightest spots in position II, the velocity of AIEgen was estimated to be around $1.4–2.2 \, \mu m \, s^{-1}$ (Supplementary Table 1). The quartz nanopore remained good capability of the continuous reversible modulation for around 30 min. In our following studies, we endeavor to largely eliminate the volatilization of the acetonitrile for enhancing long time performance and reusability of this method.

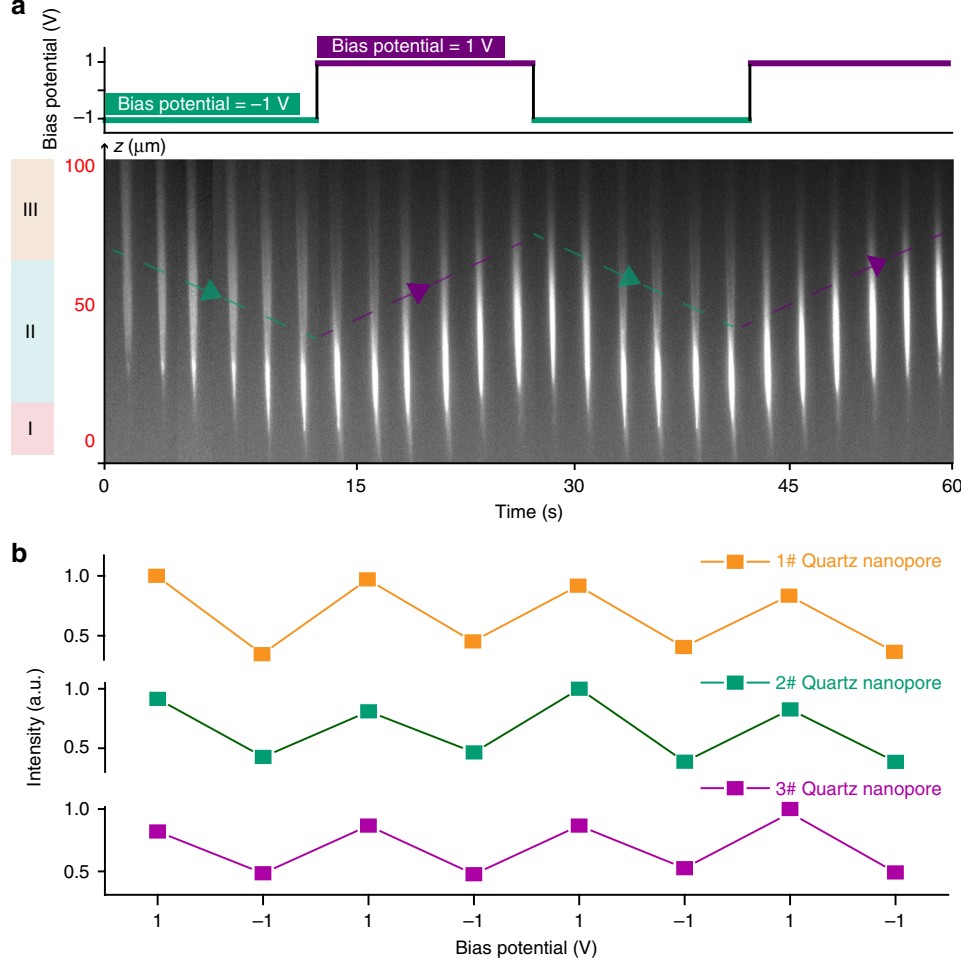

**Fig. 2** The reversible manipulating of DMTPS-DCV emission inside a nanopore. **a** The alternating applied bias potential between 1 V and −1 V inducing the reversible emission of DMTPS-DCV. The time interval for stepping the potential is 15 s. The nanopore could be divided into three areas alone with the $z$ position as confinement I, II, and III, respectively. At the confinement II in nanopore, the DMTPS-DCV exhibits the highest emission. **b** The normalized fluorescent intensity from three individual quartz nanopores at repeatable potential illustrates the consistent behavior for electrochemically manipulating the DMTPS-DCV. The normalized fluorescent intensity at each potential is recorded from the pixel at position $z = 50$ μm

**Manipulating DMTPS-DCV solution for single-cell injection.** Then, we electrochemically manipulated the DMTPS-DCV solution for a cellular injection with nanopore. Since the DMTPS-DCV shows poor solubility in aqueous solution, it can hardly across the membrane to permeate into the cell for the single-cell imaging. The manipulation and injection configuration are shown in Fig. 3a. The DMTPS-DCV solution was introduced into the nanopore by applying the bias potential of 1 V. Then, the voltage was held at 0 V at which the solution flow keeps inside at the internal of nanopore. Meanwhile, the quartz nanopore was slowly moved toward the cell from a few microns away by using a precise micromanipulator. Sequentially, the cell was punched and the DMTPS-DCV was injected quickly into the cell by stepping the bias voltage to −1 V. Afterwards, the quartz nanopore is recessed back into the solution rather than continuous remain inside the single cell. The whole injection process inside the single cell completes within 1 min. The injection of 10 μM AIE molecules at femtoliter is too few for generating a clear fluorescence image under a general dark-field microscopy with ×40 objective (NA = 0.6) (Supplementary Figure 13). Therefore, we further increased the concentration of DMTPS-DCV to 10 mM and then electrochemically implemented the cell injection. As shown in

Fig. 3b, the bias potential of −1 V drew the DMTPS-DCV aggregates back to the wide cavity, resulting in a low emission in tip region. When shifting the bias potential to 1 V, DMTPS-DCV was moved into the nanopore constriction, leading to an emissive nanopore tip (Fig. 3b). In contrast, other cells around could not be seen in the optical micrograph (Fig. 3b). Since the nanopore owns a relatively small diameter of 100 nm, it causes less damage to the cell. Note that there is no change in the cell shape, size, or position after the injection (Supplementary Figure 14), which demonstrates the excellent cell viability. Moreover, a previous study demonstrates the normal behavior of the cell and the healthy cell cycle after 1-min 100 nm nanopore punching at 1 V[33]. Therefore, the ~100 nm nanopore has a significantly lower impact on the functionality of the injected cells. By using this method, the AIEgen could be electrochemically manipulated for a direct single-cell imaging. To further incorporate with the advanced optical microscopy, this method will promote the effective AIE detection of biomarkers in complex cell environments. Since the turn-on signal in AIE detection could be easily perturbed by the ambient environment, the direct injection of the AIE molecule into the cell may reduce the undesired interference during the uptake process.

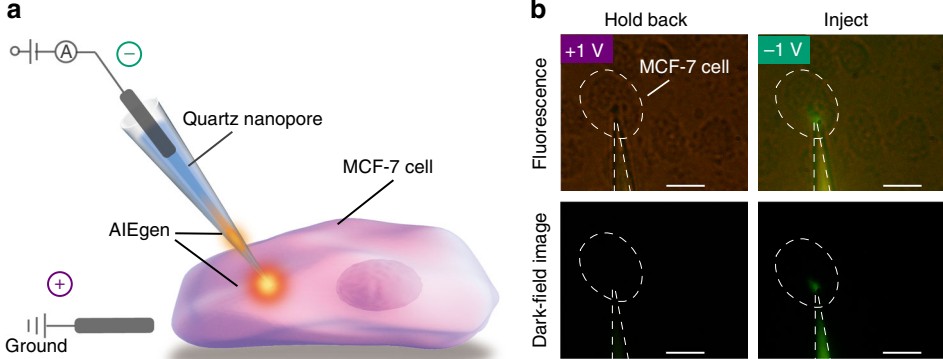

**Fig. 3** Cell injection of DMTPS-DCV using electrochemically nanopore manipulation. **a** The illustration for the cell injection. The DMTPS-DCV is sucked into the nanopore at 1 V, then the nanopore is translated toward the cell and penetrates through the cell membrane. The DMTPS-DCV is injected inside the cell at −1 V. **b** The fluorescence micrographs (upper) and dark-field images (bottom) of immobilized MCF-7 cell. Left: After quartz nanopore punches into the cell, the DMTPS-DCV is hold back away from the nanopore tip at 1 V, leading to the less emission at the nanopore tip; Right: DMTPS-DCV is electrochemically driven into the single cell at −1 V, which enlightens the intra cell. The concentration of DMTPS-DCV in this injection is 10 mM. Scale bars, 20 μm

## Discussion

The result presented here provides the direct evidence to the manipulation and visualization of dynamic AIE process in a confined nanopore, which is hardly be achieved by previous strategies. By electrochemically controlling the 26 fL DMTPS-DCV solution, the nanopore could be gradually illuminated from the constricted tip to the inside confinement at a velocity of around 1.4–2.2 μm s$^{-1}$ in a highly reversible and reproducible manner. In this regard, the AIE effect caused by the restriction of intramolecular motions could be linked to the dynamic motion of the AIEgen inside nanopore. Compared to the conventionally electrical and optical switch of the chromophores, this method could not only achieve the reversible manipulation of the AIE emission in "on-to-off" and "off-to-on" manner but also could realize the electrochemical loading of femtoliter AIEgen solution. The photoemission manipulation in our present work is facilely achieved under a moderate bias potential utilizing a very common AIEgen without special design. The nanopore method demonstrated here won't cause undesirable high-background or extra phototoxicity when delivering luminescent materials into cells. Since the AIE emission could be electrochemically controlled, it allows for a femtoliter cell injection of the AIEgen with poor cell permeability which does not need more sophisticated modifications. By further incorporating with the efficient AIEgen, our method would facilitate the delivery of femtoliter AIEgen into the single cell for the anticipated applications of high resolution single-cell imaging, single-cell diagnostics, single-cell theranostics, bacteria imaging, photodynamic therapy, and protein/DNA detection etc.. If extending this method to the charged AIEgens, one should ensure that the changes of the surface tension that drives the ingress of organic AIE solution or the expulsion of the aqueous solution should faster than the electrokinetic processes[27]. Additionally, the electrostatic adsorption of the positively charged AIEgens into the negatively-charged inner wall of nanopore may prevent one from the electrochemically modulation of the emissions. This limitation could be overcome by functionalizing the inner surface of nanopore, e.g., salinization to form a hydrophobic surface[27]. The previous study reports that the synergistically enhanced ion-gating effect induced by the outer and inner surface probe of the nanopore[34]. Therefore, the modification of nanopore will further help us to reversibly modulate the AIE emissions with the multi-stimulus, which is in response to the environmental variations of pH, temperature, force, and photon etc. Especially, the real-time visualization of the AIE emission in a nanopore could prompt us to explore the fluid mechanics inside confinement including the local pressure effects and electrokinetic effects.

In order to improve the compatible of our method to various fluorescent spectroscopy, in our lab, we are continuing to develop the specialist portable electrochemical instrument for this technique. Therefore, we expect this reversible manipulation could promote AIE to be applied in various fields from cell biology to nanolithography to optical devices in a highly reproducible manner. More importantly, the quartz nanopore provides a suitable platform for the AIE which would be used for on-site environmental monitoring and diseases diagnosis such as fluorescence image-guided surgery.

## Methods

**Electrical measurements**. A conical quartz nanopore was mounted into a PDMS flow cell with two compartments. Then, the plasma cleaner was used to clean the PDMS device. After assembly, the electrolyte solution was filled into the PDMS cells. Here, the DMTPS-DCV (10 μM) and TBAPF6 (10 mM) were added into outside acetonitrile solution while the KCl (10 mM) aqueous solution were added into inside nanopore, respectively. A pair of Ag/AgCl electrodes were used to applied the bias potential. The electrical recording were performed on Axon 200B amplifer (Molecular Devices Co., USA). The current response was filtered by a low-ass filter of 5 kHz. The analog-Digital converter (Digidata 1550 A, Molecular Devices, USA) was employed to acquire the digital signal at a sampling rate of 100 kHz. All the current response was monitored using a PC running Clampfit 10.7 software (Molecular Devices Co., USA). The electric measurements were performed at 25 °C. Cell approaching process was controlled by micromanipulator (PCS 6200, Burleigh Instrument, Fishers, NY).

**Optical detection system**. A standard dark-field microscope system was used for illuminating and detecting the process of the dynamic emission of DMTPS-DCV in the nanopore cell. The dark-field illumination is built on an inverted microscope (Ti-U Inverted Microscope, Nikon Co., Japan). The unpolarized white light from a 100 W halogen lamp is filtered at λ = 450–490 nm which is focused on the nanopore chip. Then the fluorescent light signal is collected by a ×40 objective separately (NA = 0.6) from the bottom of the flow cell. The output image from the microscope is reconstructed on the assisted digital EMCCD camera at 50 ms exposure by lenses. Therefore, the temporal resolution of imaging system should be 50 ms in capturing the images.

## Data availability

The data that support this study are available within the article and its Supplementary Information files or available from the authors upon request.

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

## Acknowledgements

This research was supported by the National Natural Science Foundation of China (21788102, 21421004) and the Fundamental Research Funds for the Central Universities (222201718001, 222201717003).

## Author contributions

Y-L. Y., H. T., and Y-T. L. designed the research; Y-L. Y., Y-J. L., J. M., R. G., and Y-X. H. performed the research; Y-L. Y., Y-J. L., and R. G. analyzed the data; and Y-L. Y., J. M., H. T., and Y-T. L. wrote the paper.

## Additional information

**Competing interests:** The authors declare no competing interests.

