## [Peer Review File · Nature Communications]

Reviewers' comments:

Reviewer #1 (Remarks to the Author):

Long, Tian and coworkers describe a new approach for dynamically manipulating the aggregation-induced emission (AIE) with revisable “on-to-off” and “off-to-on” manner. Although thousands of reported papers utilize AIE for a wide range of applications, nearly all of them focus on measuring the initial dispersed state to the ultimate aggregated state. The real-time recording the dynamic transition states between these two optical states is a long-term challenge in AIE field since it lacks appropriate experimental approach. This innovative paper presents a very smart idea which takes advantages of confined space in quartz nanopore to electrochemically manipulate the reversible conversions of AIE emission. Unlike the traditional AIE analysis in bulk solution, this method provides a revolution solution to break the limitation of AIEgens for achieving reversible sensing and reproducible optical device. The authors clearly recorded the highly repeatable movement of AIEgen along the nanopore at a controllable speed of 1.4 – 2.2 $\mu\text{m/s}$. This impressive method further applied to deliver the impermeable AIEgen into a single cell. Therefore, utilizing the nanopore confinement for reversible controlling the emission of AIEgen is a novel and great attempt in vigorous AIE fields, which facilitate AIE for its practical applications in biosensing, single cell analysis and devices. Given that, I consider this manuscript is of great importance in nanoscience and bioscience. I believe it will be of broad interest to the general readership of Nature Communications. This manuscript is well written and organized. I strongly recommend its publication in Nature Communications after a minor revision.

1. The authors discussed that “the number of DMTPS-DCV molecules in the tip of nanopore could be roughly estimated to approximately 161000.” The author should provide the detailed calculation procedures for this statement.
2. AIEgen could be electrochemically controlled by the nanopore. What is the potential distribution along the nanopore? Could the author provide more detailed description for the electric field in the tip of nanopore.
3. The author used 10 mM KCl as the aqueous electrolyte. I am wondering if the concentration of electrolyte would affect the electrochemical modulations?
4. The author set the bias potential at -1 V or +1 V to reversibly control the motion of AIEgen. The author is encouraged to perform the experiment at low potential to slow the velocity of AIEgen, which may facilitate visualize the miscible process.
5. The author should discuss the possibility for developing this method as a generalized

approaching in AIE measurements. For example, the application field, the possible AIEgen and the limitations. It will be better if this nanopore-AIE combination could be further improved as a commercial device for broad applications.

6. What is the temporal resolution of their imaging system? The author should synchronize record the electric response with the optical signals, which could further support the real-time modulation of this method.

7. Could the authors give a detailed background to demonstrate why previous reports did not realize this revisable process? Therefore, the readers could more clearly understand the preponderance of this method.

Reviewer #2 (Remarks to the Author):

“Manipulating and Visualizing the Dynamic Aggregation-Induced-Emission Confining within a Quartz Nanopore”, by Yi-Lun Ying, Yuan-Jie Li, Ju Mei, Rui Gao, Yong-Xu Hu, Yi-Tao Long*, He Tian

In the manuscript, the authors described a physical method to manipulate the photo-emission of AIEgen. Specifically, a voltage was applied to a quartz nanopore filled with AIEgen to control the confined space of the molecules and achieve the manipulation of their emission.

Experimentally, the authors observed the “nanopore-size-dependent restriction of AIEgens for the revisable conversion of “on-to-off” and “off-to-on” emission. To demonstrate the potential application, the nanopore of AIEgen was delivered into a living cell for voltage-controlled single cell imaging.

Comments:

1. Photo-emission of AIE molecule is based on the restriction of freely-rotating group(s) in the molecule. This can be achieved chemically and physically. In the study reported in this manuscript, the authors chose the later approach. Scientifically, there is no surprise to see the increase of photo-emission from AIE molecules when they are driven to a structure of more spatial restriction, the tip region of their nanopore. In my opinion, manipulation of light emission via the voltage-controlled confinement of AIE molecules in a quartz nanopore does not introduce any new concept. Rather, it makes control of light emission technically more complicated in comparison with other approaches.

2. The movement of AIE molecules in nanopore is directly determined by electrical field intensity and profile in the nanopore, not the voltage applied to the structure. The relationship between e-field and voltage is complicated and highly dependent on the geometry of electrodes

and other boundary conditions. The characterization of the dynamic AIE confining within the quartz nanopore should be based on electrical field intensity.

3. There are many ways to manipulate the photo-emission from a variety of molecules and nanoparticles, for example, those photoactivatable molecules used in super-resolution microscopy and the quantum dots of emission controlled optically/electrically. I strongly suggest that the authors compare their method with other approaches and find out whether their method indeed provides any specific advantage(s).

4. The authors demonstrated an application of their technology in biological imaging. They claimed that the AIEgen could be electrochemically manipulated for a direct single cell imaging and their method will promote the effective AIE detection of biomarkers in complex cell environments. For biological imaging, it is far from even adequate to revolve a single cell. The authors should, at least, show that their method was capable of imaging subcellular structures. More importantly, they should keep in mind that 100 nm size of a nanopore is too large to label any biomolecule or structure without changing their mobility and biological function. The trend to design biological labeling technique is to minimize the functionality change(s) induced by the labeling agent to the targeted bio-molecule or structure in cell.

Reviewer #3 (Remarks to the Author):

This manuscript, Manipulating and visualizing the dynamic aggregation-induced emission within a confined quartz nanopore, innovatively take advantage of the nanopore confinement to electrically manipulating the dynamic aggregation of AIEgens for achieving the controllable and revisable aggregation-induced emission. In conventional AIE experiments, AIEgens was subjected to poor solvent for achieving an aggregation state, thereby enabling the following emissive applications. Actually, the emission of AIEgens is hardly to be a reversible process due to the uneasy recovery of the aggregated AIEgens into the dispersion state. This obstacle puts a long-term and big challenge which impedes the reversible use of AIE sensors. Note that the reusable characteristic is very important for real applying AIEgens to achieve a robust and cost-effective sensor. To clearly address this challenge, Tian and coworkers presented the novel design of electrochemically confined nanopore together with the delicate optical-electrochemical configure in this manuscript. Upon their rational design and discussions, this manuscript gives us a very beautiful strategy for highly reversible manipulating the AIE emission. Different from the previous researches which applied nanopore for single molecule sensing (Joshua B. Edel et al. Nature Communications 2017; Aleksandar P. Ivanov et al. Nature Communications 2017, 586) and single particle trapping (Vahid Sandoghdar et al., Nature Communications 2014, 3380), the nanopore confining of AIEgen is an initiative effort to provide many unexpected inspirations for further nanopore-related researches.

Based on the above considerations, the authors presented a very comprehensive study both at the research strategy and data analysis on nanopores design and manipulation of AIEgen. Due to the

critical and prudent demonstrations, I hereby strongly recommend this work to be published in Nature Communications after address the following concerns.

1. The author presented that “AIE illuminates while moves along nanopore from the constricted tip to inside cavity at a velocity of 1.4 – 2.2 $\mu\text{m/s}$ ” in the abstract. However, the author is encouraged to provide the detailed calculation process of the velocity, even in the Supplementary Information. Therefore, the readers could follow their study.
2. The electrochemical properties of the nanopore easily affected by the charges from the pore and analyte. For example, the ion current rectification is shown in nanopores with asymmetric surface charges or shapes. The author employed the DMTPS as the model molecule for their proof-of-concept idea since this type of molecule is most widely used in AIE. The idea of employing DMTPS is good, However, I’m wondering if the charge of the inner wall of the nanopore or the charges/dipole of AIEgen could affect the dynamic emission process? It might be not easy to synthesis AIEgens, but the author should at least discuss all the possibility concerning with their new mechanisms.
3. This study focuses on the inside properties as designing the quartz nanopore. Recently, the role of outer surface probe for regulating ion gating of nanochannel is discussed, see Nature Communications 2018, 9, 40. The author should consider the outer surface effect of quartz nanopore to the manipulation of AIEgens.
4. The author employed the tip enhanced electric filed for modulating the motion of AIEs. As describe in previous research, the conical geometry of the nanopipette leads to a localized electric field, since all of the potential drop occurs in the tip region. Therefore, the understanding of the electric field is very important for electrochemically modulating the motion of AIEs flow. The author cited Ref.22 from their group, which well performed the COMSOL simulation of the nanopipette. The authors should describe electric field distribution in the experiment condition by using the COMSOL.
5. As shown in Figure 2, the author has proved that the AIEgens could be manipulated inside nanopipette. However, what about the reversible usage of a nanopipette? How long does the nanopipette persevere in an experiment? These two points are very important for demonstrating the applicability of this method.
6. The author stated that “Previous studies suggest that the application of voltage across the liquid/liquid interface changes the surface tension, previous studies suggest that the application of voltage across the liquid/liquid interface changes the surface tension.... Therefore, we applied bias potential to trigger the motion of DMTPS-DCV solution in/out of the nanopore.” They attribute the electrical motion of the AIEgen to the liquid/liquid interface changes. Is it possible that the electro-osmotic flow or electrophoretic forces affects the motion of the AIEgen?
7. Since the AIEgen visualizes the fluidic motion from acetonitrile solution to the aqueous solution. Is it possible for this technique to explore the more fluid mechanics inside nanopipettes? The author was encouraged to provide more discussions about the fluid flow inside the nanopipette.

8. The author suggests that “the quartz nanopore provides a suitable platform for the AIE which would be used for on-site environmental monitoring and diseases diagnosis such as fluorescence image-guided surgery.” Could the author provide more charming pictures for applying this strategy in AIEgens or detailed example for the applications? These discussions would help the readers in different fields to quickly get the inspiration from their novel strategy.

9. Since the nanopore confinement provides the new way to manipulate AIEgens. The author should give more details for the experimental chamber design, the optical configuration as well as the the position of electrodes.

10. As describe in Supplementary Information, “The potential is applied using a pair of Ag/AgCl electrodes. The electrical measurements were performed on a patch clamp amplifier” However, Figure 1a and 3a lacks symbol of electric supply, amperemeter and the ground definition. According to the presented figures, it is confused that if the two electrodes link to one amplifier or two dependent amplifiers. The author should link two electrodes together to one amplifier.

**Point-by-Point Response of “Manipulating and Visualizing the
Dynamic Aggregation-Induced Emission within a Confined Quartz
Nanopore” (Manuscript ID: NCOMMS-18-00885)**

Response to Reviewer 1----- Page 02-14

Response to Reviewer 2----- Page 14-23

Response to Reviewer 3----- Page 23-37

Response to Reviewer 1

Thank you very much for your constructive comments and suggestions. Our manuscript has been greatly improved according to your guidance. In this revision, we have **i)** performed the additional experiment to clearly discuss the bias voltage effects and electrolyte effects on manipulation of AIEgen; **ii)** supplied the finite element method simulation to demonstrate the electric field inside a quartz nanopore; **iii)** proposed the wide applications for this method in the Discussions. The revised parts are marked in red-color font in both the **Main text** and **Supplementary Information**. The details of the major revision are listed as follows:

***Q1:** The authors discussed that “the number of DMTPS-DCV molecules in the tip of nanopore could be roughly estimated to approximately 161000.” The author should provide the detailed calculation procedures for this statement.*

A1: Thank you very much for your suggestions. We calculate the number of DMTPS-DCV molecules according to volume of acetonitrile filled inside the nanopore (V_{tip}). According to the Figure S3 and S7, the tip volume of nanopore could be considered as a cylinder. Therefore, V_{tip} was calculated as follows:

$$V_{tip} = \frac{\pi r^2 h}{3} \quad (1)$$

Where h is the height of the acetonitrile inside nanopore, r is radius of acetonitrile part. According to the Figure S7, r is 1.49 μm which is calculated by $r = \tan \theta \times h$, where θ is the the half-cone angle of the conical-shaped area. We're sorry for the typo in Figure S7 to display the height of the acetonitrile inside nanopore which should be 15 μm . In this revise, we have corrected Figure S7 as follows. Therefore, the V_{tip} is 26 fL. Since the concentration of DMTPS-DCV in the outside acetonitrile solution is 10 μM , we assume that 26 fL of 10 μM DMTPS-DCV was driven into the tip of nanopore. Therefore, the number of DMTPS-DCV molecules in the tip of nanopore is estimated to approximately 161000.

In this revision, we have added the calculation details in Supplementary Information and corrected Figure S7 as follows:

Figure S7. Electrochemically ingress of acetonitrile into the aqueous filled quartz nanopore. (a) Initial immersion at 0 V; (b) Ingress of 26 fL acetonitrile after potential was stepped to 1 V; (c) Complete egress of acetonitrile at -1V. The 10 mM TBAPF6 and 10 mM KCl were used as organic and aqueous electrolyte to conduct the ionic flow.

“We calculate the number of DMTPS-DCV molecules according to volume of acetonitrile filled inside the nanopore (V_{tip}). According to the Figure S3 and S7, the tip volume of nanopore could be considered as a cylinder. Therefore, V_{tip} was calculated as follows

$$V_{tip} = \frac{\pi r^2 h}{3} \quad (4)$$

Where h is the height of the acetonitrile inside nanopore, r is radius of acetonitrile part. Here, h is 15 μm , r is 1.49 μm which is calculated by $r = \tan \theta \times h$, where θ is the the half-cone angle of the conical-shaped area. Therefore, the V_{tip} is 26 fL. Since the concentration of DMTPS-DCV in the outside acetonitrile solution is 10 μM , we assume that 26 fL of 10 μM DMTPS-DCV was driven into the tip of nanopore. Therefore, the number of DMTPS-DCV molecules in the tip of nanopore is estimated to approximately 161000.” (Page10, Supplementary Information)

Q2: AIEgen could be electrochemically controlled by the nanopore. What is the potential distribution along the nanopore? Could the author provide more detailed description for the electric field in the tip of nanopore.

A2: Thank you very much for your comments. To describe the potential distribution along the nanopore at room temperature of 298 K, we carried out the finite element method simulation (FEM) by coupled Poisson-Nernst-Planck (PNP) and Navier-Stokes (NS) equations. The COMSOL Multiphysics software (COMSOL Inc., Burlington, MA, USA) is used for the FEM simulation. The geometry of the model nanopore is based on the SEM image of the nanopore which are $d = 100$ nm, $\theta = 5^\circ$, $l_{\text{nanopore}} = 25$ μm (Figure 1a and S1-2). The surface charge density of the nanopore was set as -10 mC/m^2 . To accelerate the calculation, the yellow line of boundary ⑤ is defined as surface of glass with surface charge density of -10 mC/m^2 while green line of boundary ④ set as surface of glass without surface charge density. The 2D axisymmetric geometry of a nanopore for the simulation is shown in Figure S5. Note that these geometries are not drawn to scale.

Figure S5. The 2D axisymmetric geometry, the mesh for the finite-element simulation, and the boundary settings of the nanopore for the simulation of the potential distribution.

Blue lines (①, ②, ③, ⑥, ⑦, ⑧) are the internal bulk solution and external bulk solution. Green line of ④ is the surface of glass. Yellow line of ⑤ is the surface charge of glass.

In order to simplify the simulation, here, both the outside and inside solution are set as 10 mM KCl. The dynamics of the ions in the channel is governed by the Poisson-Nernst-Planck (PNP) equations which relate surface charge with ionic fluxes and corresponding conductivity distribution within a glass nanopore. The ionic flow is computed by the Nernst-Planck (NP) equation (eq. 1) where the diffusion, migration, and convection terms are included.

$$\mathbf{J}_i = -D_i \nabla c_i - \frac{Fz_i}{RT} D_i c_i \nabla \Phi + c_i \mathbf{u} \quad (1)$$

In equation (eq. 1), \mathbf{J}_i is the ionic flow vector, F is the Faraday's constant, T is the absolute temperature, Φ is the potential, \mathbf{u} is the position-dependent fluid velocity, respectively. D_i , C_i , and Z_i represent the diffusion coefficient, the concentration, and the charge of species i in solution, respectively.

As neither species are transported into or out of the surface of glass (e.g. boundaries ②, ③, ④, ⑤, ⑥ and ⑦ in 2D axial symmetric model, Figure S5), there is no normal flux; the boundary condition is set as:

$$\mathbf{N} \cdot \mathbf{J}_i = 0$$

The relationship between the electric potential and ion concentration is described by Poisson equation, eq. 2.

$$\nabla^2 \Phi = -\frac{F}{\epsilon} \sum_i z_i c_i \quad (2)$$

Where ϵ is the dielectric constant of the medium.

The flow distribution is given by the Navier-Stokes equation, eq. 3, describing the pressure and electrical force driven flow.

$$\rho \nabla \cdot \mathbf{u} = -\nabla p + \eta \nabla^2 \mathbf{u} - F (\sum_i \sigma_i c_i) \nabla \Phi \quad (3)$$

Here, ρ and η are the density and viscosity of the fluid, and p is the pressure.

p sets as $1 \times 10^{-3} \text{ kg/m}^3$ and η sets as $1 \times 10^{-3} \text{ Pa} \cdot \text{s}$.

Numerical solution of the equations involves discretization, which uses a mesh. Representative meshes for the finite-element simulations are shown in Figure S5. In each model, right part shows the entirety of the domain and right part illustrates the region at the tip of nanopore. The finite element simulations were carried out with COMSOL Multiphysics 5.2a (COMSOL Inc., Burlington, MA, USA) operated on a Lenovo P500 workstation (Intel(R) Xeon(R) CPU E5-1620 v3@3.50GHz, 4core, 32GB RAM)

Therefore, the simulation results in Figure S6 shows the electrical potential distribution along the centerline axis of a 100 nm diameter nanopore with a surface charge -10 mC/m^2 for applied biases potential at -1 V and 1 V . The FEM demonstrates that the largest fraction of the bias potential drops at the narrowest tip of nanopore with approximately region of $15 \mu\text{m}$ in length.

Figure S6. Electric potential along the center-line z axis ($r = 0$) of a 100 nm diameter nanopore in 10 mM KCl as a function of applied bias (ΔV) across the nanopore at -1 V and $+1 \text{ V}$, respectively.

In this revision, the simulation results are supplied in Main Text as follows:

“The finite element method simulation (FEM)^{19,31} was used to demonstrate that the largest fraction of the bias potential drops at the narrowest tip of nanopore with

approximately region of 15 μm in length (Figure S5-S6), which induced a strong electric field at the tip confinement for generating the electrochemical surface tension effects. The detailed description of the FEM simulation is shown in Supplementary Information.”
(Page 6, Line 3-8, Main Text)

The description of the FEM simulation has been supplied in Page 7-9, Supplementary Information.

Q3: The author used 10 mM KCl as the aqueous electrolyte. I am wondering if the concentration of electrolyte would affect the electrochemical modulations?

A3: The physicochemical processes involved in the electrochemical modulation of AIEgens are based on the electrowetting. The liquid flow is largely determined by the surface tension effects. The concentration of aqueous electrolyte will affect the liquid flow of AIEgen as described in previous study (*Proc. Natl. Acad. Sci.*, 2007, 104, 11895-11900). In our experiments, as the electrolyte concentration down to 1 mM (Figure S8), the applied voltage drops within the bulk solution phase rather than across the liquid/liquid interface. Therefore, it is hard to achieve the electrochemically controlling the surface tension and the related fluidic motion. As the electrolyte concentration increase to 100 mM (Figure S9), the significant decreasing of the pipette resistance prevents the electrochemical control of the fluid motion. Therefore, in this experiment, we used 10 mM tetra-*n*-butylammonium and 10 mM KCl were used here as electrolyte in outside and inside solution.

The supplied experimental results and discussions have been added in this revision as follows:

“Since the AIEgen flow is largely determined by the surface tension effects in the confinement, the extremely high or low KCl concentration prevent the electrochemical control of the fluid motion (Figure S8-9). Here, the optimized the KCl concentration was set as 10 mM.” (Page 6, Line 13-16, Main Text)

“The liquid flow is largely determined by the surface tension effects. The concentration of aqueous electrolyte will affect the liquid flow of AIEgen as described in previous study³. As the aqueous electrolyte concentration down to 1 mM (Figure S8), the applied voltage drops within the bulk solution phase rather than across the liquid/liquid interface. Therefore, the it is hard to achieve the electrochemically controlling the surface tension and the related fluidic motion. As the electrolyte concentration increases to 100 mM (Figure S9), the significant decreasing of the nanopore resistance hinders the electrochemical control of the fluid motion. Therefore, 10 mM KCl were used here as electrolyte in inside aqueous solution to efficient ingress of the acetonitrile solution.

Figure S8. Electrochemically ingress of acetonitrile into the aqueous filled quartz nanopore at the KCl concentration of 1 mM. (a) Initial immersion at 0 V; (b) The bias potential was stepped to 1 V.

Figure S9. Electrochemically ingress of acetonitrile into the aqueous filled quartz

nanopore at the KCl concentration of 100 mM. (a) Initial immersion at 0 V; (b) The bias potential was stepped to 1 V.” (Page 10-11, Supplementary Information)

Q4: The author set the bias potential at -1 V or +1 V to reversibly control the motion of AIEgen. The author is encouraged to perform the experiment at low potential to slow the velocity of AIEgen, which may facilitate visualize the miscible process.

A4: We have supplied the voltage dependent experiment in this revision. As shown in the Figure S10, the insufficient electric field at low bias potential (absolute value < 0.5 V) could not induce the strong surface tension for the organic phase movements. As the electric field increase, the acetonitrile solution ingress/egress significantly which generates significant movement along the z position of the nanopore. Therefore, we optimized the bias potential as ± 1 V in our experiments for manipulating the AIE motions. The experimental results for the dependence of the bias potential have been added in both Main Text and Supplementary Information as follows:

“As setting the optimal bias potential to 1 V (Figure S10), the acetonitrile solution of DMTPS-DCV is driven into the inside aqueous solution from the exterior.” (Line 17-19, Page 6, Main Text)

Figure S10. Dependence of the bias potential on the electrochemically ingress of acetonitrile into the aqueous filled quartz nanopore at the KCl concentration of 10 mM.

The tip of the nanopore defines as z position of 0 μm . Therefore, the ingress of the acetonitrile produces the positive z position value while the egress of the acetonitrile generates the negative z position value. Since ± 1 V triggers the the acetonitrile moving deepest inside the nanopore, we set the bias potential as ± 1 V in the following AIE manipulations.” (Page 12, Supplementary Information)

Q5: The author should discuss the possibility for developing this method as a generalized approaching in AIE measurements. For example, the application field, the possible AIEgen and the limitations. It will be better if this nanopore-AIE combination could be further improved as a commercial device for broad applications.

A5: Thank you very much for your suggestions. In this proof-of-concept manuscript, we employed DMTPS-DCV as a model molecule to show the electrochemically reversible modulation of its emission. In this design, the organic solution of AIEgens was added into the outside of nanopore whereas the inside was filled with the aqueous solution. Under the applied bias potential, the motion of AIEgens could be triggered to move in/out of the nanopore. Therefore, this method could control the organic soluble AIEgens to electrochemically behavior “on-to-off” and “off-to-on” emission. By using this method, the emission of organic soluble AIEgens could be manipulated to achieve cell imaging, bacteria imaging, photodynamic therapy, protein/DNA detection and etc.. The possible molecule could include but not limited to derivatives of tetraphenylethene, hexaphenylsilole, diphenylethylene anthracene and diphenyl fluorene. In order to improve the compatible of our method to various fluorescent spectroscopy, in our lab, we are continuing to develop the portable electrochemical instrument. Therefore, the commercial device based on nanopore-AIE method could be expected which will be applied in wide AIE applications.

In this revision, we have added the following statements to describe the prospect of this method in Main Text:

“By further incorporating with the efficient AIEgen, our method would facilitate the

delivery of femtoliter AIEgen into the single cell for the anticipated applications of high resolution single cell imaging, single cell theranostic, single cell diagnostics, bacteria imaging, photodynamic therapy, protein/DNA detection and etc..” (Page 11, Line 23-25 & Page 12, Line 1-2, Main Text)

“The previous study reports that the synergistically enhanced ion-gating effect induced by the outer and inner surface probe of the nanopore.³³ Therefore, the modification of nanopore will further help us to reversibly modulate the AIE emissions with the multi-stimulus which is in response to the environmental variations of pH, temperature, force, photon and etc. Especially, the real-time visualization of the AIE emission in a nanopore could help us to explore the fluid mechanics inside confinement including the local pressure effects and electrokinetic effects.” (Page 12, Line 7-12, Main Text)

Q6: *What is the temporal resolution of their imaging system? The author should synchronize record the electric response with the optical signals, which could further support the real-time modulation of this method.*

A6: Thank you very much for your comments. As present in our Supplementary Information, the output image from the microscope is reconstructed on the assisted digital EMCCD camera at 50 ms exposure by lens. Therefore, the temporal resolution of imaging system should be 50 ms in capturing the images. Here, we synchronized recorded the optical signals with the electric response in all the experiments, leading to a real-time modulation of this method. In this revision, we have supplied a screening recording video to demonstrate the synchronized recording (Part II: Synchronized Optical and Electrical Recording, Supplementary Video). As shown in the Supplementary Video, the ingress of the outside DMTPS-DCV at 1 V leads to an increase of the interface resistance, which exhibits a decrease of the ionic current value. Inversely, the out flow of the DMTPS-DCV solution from the inside nanopore to the outside nanopore generates a decreasing of the interface resistance, which suddenly increases the value of ionic current before gradually reach to a constant

value.

In this revision, we have supplied the following detailed description of our recording process as well as a supplementary video:

“Detailed Schematic configuration for synchronized collection of electrical and optical signals are shown in Figure S2.” (Page 5, Line 4-5, Main Text)

“Note that the electric response was synchronized recorded with the optical signals in all the experiments. As shown in the Supplementary Video, the ingress of the outside DMTPS-DCV at 1 V leads to an increase of the interface resistance, which exhibits a decrease of the ionic current value. Inversely, the out flow of the DMTPS-DCV solution from the inside nanopore to the outside nanopore at -1V generates a decreasing of the interface resistance, which suddenly increases the value of ionic current before gradually reach to a constant value.” (Page 7, Line 22-25 & Page 8, Line 1-2, Main Text)

“Therefore, the temporal resolution of imaging system should be 50 ms in capturing the images.” (Page 3, Line 8-9, Supplementary Information)

Figure S2. Schematic configuration for integrating the inverted fluorescence microscopy with the ultralow ionic current acquisition system. The synchronized collection of electrical and optical signals was guaranteed by the multichannel design of the analog to digital converter (ADC), then monitored by the computer. (*Page 4, Supplementary Information*)

Q7: *Could the authors give a detailed background to demonstrate why previous reports did not realize this revisable process? Therefore, the readers could more clearly understand the preponderance of this method.*

A7: Thank you very much for your concerns. Many researchers keen on this kind of charming molecule for developing wide range of applications including bio/chemosensing, bioimaging and optoelectronic devices. Although AIEgens has been confined into the porous organic frameworks (*Chem. Mater.*, 2016, 28, 7889), two-dimensional (2D) nanomaterials (*J. Am. Chem. Soc.*, 2018, 140, 4035), confined chiral nanotubes (*Adv. Mat.* 2017, 29, 1606503) and nanochannels of metal-organic frameworks (*Nanoscale*, 2016,8, 17489), these physical restrictions only induce the emission of AIEgens rather than the reversible manipulation of the emission in “on-to-off” and “off-to-on” manner. Therefore, it is still a big challenge in reversible manipulating the emission of AIEgens from the initial dispersed state to the ultimate aggregated state through either chemical or physical approaches. This drawback puts an impenetrable barrier for AIE to achieve reversible sensing and reproducible optical device. One of the potential solutions is to reversibly and dynamically manipulate the emission of AIEgens. However, this concept is hardly achieved by all of the previous AIE approaches and strategies although it has been studied almost 20 years. To overcome the above challenge, in this study, we take the advantage of nanopore confinement to propose a new strategy of AIE which is nanopore-length-dependent emission of AIEgens. By electrochemically manipulating nearly 161000 AIEgens inside nanopore confinement, we for the first time achieve the dynamical manipulation and visualization of AIE emissions with the reversible “on-to-off” and “off-to-on” manner. We further applied this dynamic manipulation to achieve a target

delivery of AIEgens into a single cell, which opens up the new possibility to design the powerful and practical AIE platform.

In this revision, we have supplied clear statements to clearly describe the aim of this work. The revised statements are listed as follows:

“Although AIEgens has been confined into the porous organic frameworks ¹⁴, two-dimensional (2D) nanomaterials ¹⁵, confined chiral nanotubes ¹⁶ and nanochannels of metal-organic frameworks ¹⁷, these physical restrictions only induce the emission of AIEgens rather than the reversible manipulation of the emission in “on-to-off” and “off-to-on” manner.” *(Page 3, Line 18-21, Main Text)*

“More importantly, the nanopore technology achieves electrochemically manipulate the molecules or solutions inside confinement.²⁶⁻²⁷ Therefore, confining the AIEgens into the nanopore enables the numbers of AIE molecules under study to be electrochemically tuned and controlled for aggregation in ways not possible with bulk system.” *(Page 4, Line 2-6, Main Text)*

Response to Reviewer 2

Thank you very much for your constructive comments and suggestions, which guided us in improving our manuscript. To elaborate the importance of our manuscript, we have revised the Conclusion and Introduction for highlighting the novelty of our nanopore-based reversible AIE emissions, which shows the promising applications ranging from single cell imaging, diagnostics and to the theranostics. The mechanism for the electric field controlling of AIEgen motion is further supported by the finite element method simulations, which shows that the above 90% electric potential drops at the tip of nanopore leads to the electrochemical surface tension effects. The achievements of our manuscript merely open a crack of the entrance door towards the reversible controlling of the AIE emission and will drive this technology to a wider application scope.

Q1: Photo-emission of AIE molecule is based on the restriction of freely-rotating group(s) in the molecule. This can be achieved chemically and physically. In the study reported in this manuscript, the authors chose the later approach. Scientifically, there is no surprise to see the increase of photo-emission from AIE molecules when they are driven to a structure of more spatial restriction, the tip region of their nanopore. In my opinion, manipulation of light emission via the voltage-controlled confinement of AIE molecules in a quartz nanopore does not introduce any new concept. Rather, it makes control of light emission technically more complicated in comparison with other approaches.

A1: Thank you very much for your concerns. After proposed in 2001, the aggregation-induced emission fluorogen (AIEgen) have attracted intensive attentions from Chemistry, Material Science, Biotechnology and Nanotechnology due to its features of high emission efficiency in the aggregated state. Many researchers keen on this kind of charming molecule for developing wide range of applications including bio/chemosensing, bioimaging and optoelectronic devices. Although AIEgens has been confined into the porous organic frameworks (*Chem. Mater.*, 2016, 28, 7889),

two-dimensional (2D) nanomaterials (*J. Am. Chem. Soc.*, 2018, 140, 4035), confined chiral nanotubes (*Adv. Mat.* 2017, 29, 1606503) and nanochannels of metal-organic frameworks (*Nanoscale*, 2016,8, 17489), these physical restrictions only induce the emission of AIEgens rather than the reversible manipulation of the emission in “on-to-off” and “off-to-on” manner. Therefore, it is still a big challenge in reversible manipulating the emission of AIEgens from the initial dispersed state to the ultimate aggregated state through either chemical or physical approaches. This drawback puts an impenetrable barrier for AIE to achieve reversible sensing and reproducible optical device. One of the potential solutions is to reversibly and dynamically manipulate the emission of AIEgens. However, this concept is hardly achieved by all of the previous AIE approaches and strategies although it has been studied almost 20 years. To overcome the above challenge, in this study, we take the advantage of nanopore confinement to propose a new strategy of AIE which is nanopore-length-dependent emission of AIEgens. By electrochemically manipulating nearly 161000 AIEgens inside nanopore confinement, we for the first time achieve the dynamical manipulation and visualization of AIE emissions with the reversible “on-to-off” and “off-to-on” manner. We further applied this dynamic manipulation to achieve a target delivery of AIEgens into a single cell, which opens up the new possibility to design the powerful and practical AIE platform.

In this revision, we have supplied clear statements to clearly describe the aim of this work. The revised statements are listed as follows:

“More importantly, the nanopore technology achieves electrochemically manipulate the molecules or solutions inside confinement.²⁶⁻²⁷ Therefore, confining the AIEgens into the nanopore enables the numbers of AIE molecules under study to be electrochemically tuned and controlled for aggregation in ways not possible with bulk system.” (*Page 4, Line 4-8, Main Text*)

Q2: *The movement of AIE molecules in nanopore is directly determined by electrical field intensity and profile in the nanopore, not the voltage applied to the structure. The*

relationship between e-field and voltage is complicated and highly dependent on the geometry of electrodes and other boundary conditions. The characterization of the dynamic AIE confining within the quartz nanopore should be based on electrical field intensity.

A2: Thank you very much for your comments. The reversible motions of AIE molecules are controlled by the electrical field intensity and profile inside nanopore. In our design, the organic solution of AIEgens was added into the outside of nanopore whereas the inside was filled with the aqueous solution. The physicochemical processes involved in the electrochemical modulation of AIEgens are based on the electrowetting. The liquid flow is largely determined by the surface tension effects which is induced by the significant drop in electrical field at the nanopore tip. To describe the potential distribution along the nanopore at room temperature of 298 K, we carried out the finite element method simulation (FEM) by coupled Poisson-Nernst-Planck (PNP) and Navier-Stokes (NS) equations. The COMSOL Multiphysics software (COMSOL Inc., Burlington, MA, USA) is used for the FEM simulation. The geometry of the model nanopore is based on the SEM image of the nanopore which are $d = 100$ nm, $\theta = 5^\circ$, $l_{\text{nanopore}} = 25$ μm (Figure 1a and S1-2). The surface charge density of the nanopore was set as -10 mC/m². To accelerate the calculation, the yellow line of boundary ⑤ is defined as surface of glass with surface charge density of -10 mC/m² while green line of boundary ④ set as surface of glass without surface charge density. The 2D axisymmetric geometry of a nanopore for the simulation is shown in Figure S5. Note that these geometries are not drawn to scale.

Figure S5. The 2D axisymmetric geometry, the mesh for the finite-element simulation, and the boundary settings of the nanopore for the simulation of the potential distribution. Blue lines (①, ②, ③, ⑥, ⑦, ⑧) are the internal bulk solution and external bulk solution. Green line of ④ is the surface of glass. Yellow line of ⑤ is the surface charge of glass.

In order to simplify the simulation, here, both the outside and inside solution are set as 10 mM KCl. The dynamics of the ions in the channel is governed by the Poisson-Nernst-Planck (PNP) equations which relate surface charge with ionic fluxes and corresponding conductivity distribution within a glass nanopore. The ionic flow is computed by the Nernst-Planck (NP) equation (eq. 1) where the diffusion, migration, and convection terms are included.

$$\mathbf{J}_i = -D_i \nabla c_i - \frac{F Z_i}{RT} D_i c_i \nabla \Phi + c_i \mathbf{u} \quad (1)$$

In equation (eq. 1), \mathbf{J}_i is the ionic flow vector, F is the Faraday's constant, T is the absolute temperature, Φ is the potential, \mathbf{u} is the position-dependent fluid velocity, respectively. D_i , C_i , and Z_i represent the diffusion coefficient, the concentration, and the charge of species i in solution, respectively.

As neither species are transported into or out of the surface of glass (e.g. boundaries ②, ③, ④, ⑤, ⑥ and ⑦) in 2D axial symmetric model, Figure S5), there is no normal flux; the boundary condition is set as:

$$N \cdot J_i = 0$$

The relationship between the electric potential and ion concentration is described by Poisson equation, eq. 2.

$$\nabla^2 \Phi = -\frac{F}{\epsilon} \sum_i z_i c_i \quad (2)$$

Where ϵ is the dielectric constant of the medium.

The flow distribution is given by the Navier-Stokes equation, eq. 3, describing the pressure and electrical force driven flow.

$$\mathbf{u} \nabla \mathbf{u} = \frac{1}{\rho} (-\nabla p + \eta \nabla^2 \mathbf{u} - F (\sum_i \sigma_i c_i) \nabla \Phi) \quad (3)$$

Here, ρ and η are the density and viscosity of the fluid, and p is the pressure. p sets as $1 \times 10^{-3} \text{ kg/m}^3$ and η sets as $1 \times 10^{-3} \text{ Pa} \cdot \text{s}$.

Numerical solution of the equations involves discretization, which uses a mesh. Representative meshes for the finite-element simulations are shown in Figure S5. In each model, right part shows the entirety of the domain and left part illustrates the region at the tip of nanopore. The finite element simulations were carried out with COMSOL Multiphysics 5.2a (COMSOL Inc., Burlington, MA, USA) operated on a Lenovo P500 workstation (Intel(R) Xeon(R) CPU E5-1620 v3@3.50GHz, 4core, 32GB RAM)

Therefore, the simulation results exhibited in Figure S6 shows the electrical potential

distribution along the centerline axis of a 100 nm diameter nanopore with a surface charge -10 mC/m^2 for applied biases potential at -1 V and 1 V . The FEM demonstrates that the largest fraction of the bias potential drops at the narrowest tip of nanopore with approximately region of $15 \mu\text{m}$ in length.

Figure S6. Electric potential along the center-line z axis ($r = 0$) of a 100 nm diameter nanopore in 10 mM KCl as a function of applied bias (ΔV) across the nanopore at -1 V and $+1 \text{ V}$, respectively.

In this revision, the simulation results are supplied in the Main Text as follows:

“The finite element method simulation (FEM)^{19,31} was used to demonstrate that the largest fraction of the bias potential drops at the narrowest tip of nanopore with approximately region of $15 \mu\text{m}$ in length (Figure S5-S6), which induced a strong electric field at the tip confinement for generating the electrochemical surface tension effects. The detailed description of the FEM simulation is shown in Supplementary Information.”
(Page 6, Line 3-8, Main Text)

The description of the FEM simulation has been supplied in Page 7-9, Supplementary Information.

Q3: *There are many ways to manipulate the photo-emission from a variety of molecules and nanoparticles, for example, those photoactivatable molecules used in super-resolution microscopy and the quantum dots of emission controlled optically/electrically. I strongly suggest that the authors compare their method with other approaches and find out whether their method indeed provides any specific advantage(s).*

A3: Thank you very much for your concerns. In super-resolution microscopy as single-molecule localization microscopy, those photoactivatable molecules and the quantum dots usually switched between their bright and dark states. These optical controls require high laser load to achieve high enough off/on ratio to enable single-molecule localization for super-resolution microscopy thus resulting in unexpected background, more phototoxicity and complicated operations (*Photochem. Photobiol. Sci.* 2011, 10, 499). Moreover, the photoactivatable molecules and the quantum dots are always specially designed to pursue the photoactivatability. More importantly, most of these chromophores demonstrate poor cell permeability and require external additives, such as thiols (β -mercaptoethylamine, 2-mercaptoethanol, glutathione), ascorbic acid, and oxygen-scavenging agents, to efficiently blink (*Nat. Methods* 2011, 8, 1027; *Nat. Chem.* 2014, 6, 681). Therefore, these molecules and nanoparticles are hardly to be directly used in the single-cell imaging and meanwhile, it is difficult to physically manipulate their movements during the photo-switching processes. In sharp contrast, the photoemission manipulation in our present work is facilely achieved under moderate bias potential utilizing a very common AIEgen without special design. The nanopore method demonstrated here won't cause undesirable high-background or extra phototoxicity when delivering luminescent materials into cells. In this manuscript, we electrochemically manipulate the AIEgen emission by well controlling their motion inside a confined nanopore. Our manuscript showed that this method could not only achieve the reversible manipulation of the emission in "on-to-off" and "off-to-on" manner but also could realize the electrochemical loading of femtoliter AIEgen solution. More importantly, the

conventional fluorophores featured with π -planar structures usually suffer from serious self-quenching in the aggregated state (ACQ), and poor photostability. In contrast to ACQ fluorophores, aggregation-induced emission fluorogens (AIEgens) are nonemissive in dilute solution but emit intensely upon aggregate formation (*Chem. Commun.*, 2001, 1740.1741; *J. Am. Chem. Soc.*, 2002, 124, 14410–14415; *Angew. Chem. Int. Ed.*, 2015, 54, 3912–3916; *Nat. Commun.*, 2017, 8, 15639). Through simple modulation of their structures, the AIEgens can be easily tuned to satisfy certain sensing purposes with advantages of excellent signal-to-noise ratio, strong photostability, large Stokes' shift, and high portability. These three advantages provide the promising applications of this method not limited to the cell imaging, but also for the single-cell diagnostics and theranostics.

In this revision, we have added the corresponding discussions in the Main Text as follows:

“Compared to the conventional electrical and optical switch of the chromophores, this method could not only achieve the reversible manipulation of the AIE emission in “on-to-off” and “off-to-on” manner but also could realize the electrochemical loading of femtoliter AIEgen solution. The photoemission manipulation in our present work is facilely achieved under a moderate bias potential utilizing a very common AIEgen without special design. The nanopore method demonstrated here won't cause undesirable high-background or extra phototoxicity when delivering luminescent materials into cells.” (*Page 11, Line 15-21, Main Text*)

Q4: *The authors demonstrated an application of their technology in biological imaging. They claimed that the AIEgen could be electrochemically manipulated for a direct single cell imaging and their method will promote the effective AIE detection of biomarkers in complex cell environments. For biological imaging, it is far from even adequate to revolve a single cell. The authors should, at least, show that their method was capable of imaging subcellular structures. More importantly, they should keep in mind that 100 nm size of a nanopore is too large to label any biomolecule or structure*

without changing their mobility and biological function. The trend to design biological labeling technique is to minimize the functionality change(s) induced by the labeling agent to the targeted bio-molecule or structure in cell.

A4: The aim of this manuscript is to electrochemically manipulate and visualize the dynamic AIE emission by using a confined quartz nanopore. Due to the advantage of controllable motion of AIEgen, we applied this method for the direct AIEgen delivery into the living cell as a first attempt for this method. The capable of imaging subcellular structures is a long-term challenge in the optical imaging field, which is highly relied on the molecular design of chromophore. However, this is not the topic of this work. Potentially, the cell cultivation using luminescent agents with high concentration was an essential step for the cell imaging which may put a higher risk of cell toxicity. Moreover, not every luminescent agent can enter and light up the cell, because of the poor cell permeability. AIEgens have been proven high-performance cell-imaging contrast agents for their high luminosity, superior photostability, and high contrast, etc. By further incorporating with the efficient AIEgen, our method would facilitate the delivery of femtoliter AIEgen into a single cell for the high resolution cell imaging with less damage and good portability. The labelling of the biomolecule or structure in the single cell could be realized by the further designed AIEgens, but is not the scope of this work. The small diameter of our nanopore (~100 nm) causes less damage to the single cell since it achieves the multiple penetration into an individual cell according to our previous study (*J. Am. Chem. Soc.*, 2018, DOI: 10.1021/jacs.7b12106). Furthermore, our method does not depend on the cell permeability of the studied luminescent agents. Therefore, our method provides the anticipated facile applications of AIEgens for single-cell imaging, single-cell diagnostics and single-cell theranostics.

In this revision, we have added the following discussions to clearly present the anticipated applications of our methods.

“Since the AIE emission could be electrochemically controlled, it allows for a femtoliter cell injection of the AIEgen with poor cell permeability, which does not

need further sophisticated modification. By further incorporating with the efficient AIEgen, our method would facilitate the delivery of femtoliter AIEgen into the single cell for the anticipated applications of high resolution single-cell imaging, single-cell diagnostics, single-cell theranostics, bacteria imaging, photodynamic therapy, protein/DNA detection and etc..” *(Page 11, Line 22-25 & Page 12, Line 1-2, Main Text)*

Response to Reviewer 3

Thank you very much for your constructive comments and suggestions, which greatly help us to improve manuscript. We have made strong efforts to improve our manuscript from following four aspects: **i)** Including the detailed description and related figures for the simulations. **ii)** Clarifying the electrokinetic process involving in the manipulating of the AIE fluidic motion. **iii)** Describing the detailed outlook and limitation of this emerging methods.

The revised parts are marked in red-color font in both the Main text and Supplementary Information. The details of the revision are listed as follows:

Q1: *The author presented that “AIE illuminates while moves along nanopore from the constricted tip to inside cavity at a velocity of 1.4 – 2.2 $\mu\text{m/s}$ ” in the abstract. However, the author is encouraged to provide the detailed calculation process of the velocity, even in the Supplementary Information. Therefore, the readers could follow their study.*

A1: Thank you very much for your comments. The velocity of AIEgen was estimated by tracing the motion of brightest spots in position II of Figure 2 in the time interval of 15 s, which yields the value of 1.4 -2.2 $\mu\text{m/s}$. In this revision, we have supplied a table (*Table S1, Page 15, Supplementary Information*) to display the z-position of the brightest spots during the electrochemically reversible manipulating of DMTPS-DCV emission inside a nanopore confinement.

Table S1. The Z Position of the brightest spot in Figure 2a. The velocity of the movement is calculated based on the movement during the recording time interval of 15 s.

Bias Potential (V)	Z Position of the brightest spot (μm)	Recording time (s)	Velocity of the Movement ($\mu\text{m/s}$)
-1	31	0	N.A.

-1	10	15	1.4
1	43	30	2.2
-1	15	45	1.8
1	46	60	2.1

Q2: *The electrochemical properties of the nanopore easily affected by the charges from the pore and analyte. For example, the ion current rectification is shown in nanopores with asymmetric surface charges or shapes. The author employed the DMTPS as the model molecule for their proof-of-concept idea since this type of molecule is most widely used in AIE. The idea of employing DMTPS is good, However, I'm wondering if the charge of the inner wall of the nanopore or the charges/dipole of AIEgen could affect the dynamic emission process? It might be not easy to synthesis AIEgens, but the author should at least discuss all the possibility concerning with their new mechanisms.*

A2: Thank you very much for your comments. In this proof-of-concept manuscript, we employed DMTPS-DCV as a model molecule to show the electrochemically reversible modulation of its emission. In this design, the organic solution of AIEgens was added into the outside of nanopore whereas the inside was filled with the aqueous solution. The physicochemical processes involved in the electrochemical modulation of AIEgens are based on the electrowetting. The liquid flow is largely determined by the surface tension effects. However, it also depends on the water/glass and organic solvent/glass boundaries, and a three-phase water/organic/glass boundary (*J. Phys. Condens. Matter.*, 2005, 17, R705-R744; *Anal. Chem.* 2015, 87, 9070-9097; *Anal. Chem.* 2010, 82, 84-90.). The properties of the inner nanopipette wall can be changed by modifying its surface (*Nat. Commun.* 2018, 9, 40). Rendering the glass surface more hydrophobic, e.g., via salinization, could inhibits the ingress of water into the capillary. In this way, both the extent of control over the solution ingress/egress and the magnitude of the required voltage can be varied.

As describe in previous studies (*J. Phys.: Condens. Matter* **2005**, 17, R705-R774;

Natl. Acad. Sci. U. S. A. **2007**, *104*, 11895-900), the relatively high concentration of the electrolyte both in organic and aqueous solution could cross the liquid/liquid interface. For an optimal syringe performance, the ion-transfer current should be minimized by using the hydrophobic organic electrolyte. On the other hand, the electrokinetic including the electrophoresis and electroosmosis occur within the confinement of the nanopore. If applying this method to the charged AIEgens nanopipette, one should ensure that the changes of the surface tension which drives the ingress of organic solution or the expulsion of the aqueous solution should be faster than the electrokinetic processes. Furthermore, the electrostatic adsorption of the positively charged AIEgens into the negatively-charged inner wall of nanopore may also prevent one from the electrochemically modulation of the emissions.

In this revision, we have supplied the discussions for the effects from charge of the inner wall of the nanopore and the charges of AIEgen as follows.

“If extending this method to the charged AIEgens, one should ensure that the changes of the surface tension which drives the ingress of organic AIE solution or the expulsion of the aqueous solution should be faster than the electrokinetic processes²⁷. Additionally, the electrostatic adsorption of the positively charged AIEgens into the negatively-charged inner wall of nanopore may prevent one from the electrochemically modulation of the emissions. This limitation could be overcome by functionalizing the inner surface of nanopore, e.g. salinization to form a hydrophobic surface²⁷. ” (*Page 12, Line 2-7, Main Text*)

Q3: *This study focuses on the inside properties as designing the quartz nanopore. Recently, the role of outer surface probe for regulating ion gating of nanochannel is discussed, see Nature Communications 2018, 9, 40. The author should consider the outer surface effect of quartz nanopore to the manipulation of AIEgens.*

A3: Thank you very much for this important reference. This reference delights us for the further multi-stimulus of nanopore-based AIE device. In this revision, this reference has been cited as Reference 33 and we added the following statements in

the Main Text as follows:

“The previous study reports that the synergistically enhanced ion-gating effect induced by the outer and inner surface probe of the nanopore.³³ Therefore, the modification of nanopore will further help us to reversibly modulate the AIE emissions with the multi-stimulus which is in response to the environmental variations of pH, temperature, force, photon and etc. ” (Page 12, Line 8-11, Main Text)

References:

33. Li, X., Zhai, T., Gao, P., Cheng, H., Hou, R., Lou, X. & Xia, F. Role of outer surface probes for regulating ion gating of nanochannels. *Nat. Commun.* **9**, 40, doi:10.1038/s41467-017-02447-7 (2018).

Q4: The author employed the tip enhanced electric field for modulating the motion of AIEs. As describe in previous research, the conical geometry of the nanopipette leads to a localized electric field, since all of the potential drop occurs in the tip region. Therefore, the understanding of the electric field is very important for electrochemically modulating the motion of AIEs flow. The author cited Ref.22 from their group, which well performed the COMSOL simulation of the nanopipette. The authors should describe electric field distribution in the experiment condition by using the COMSOL.

A4: Thank you very much for your suggestions. To describe the potential distribution along the nanopore at room temperature of 298 K, we carried out the finite element method simulation (FEM) by coupled Poisson-Nernst-Planck (PNP) and Navier-Stokes (NS) equations. The COMSOL Multiphysics software (COMSOL Inc., Burlington, MA, USA) is used for the FEM simulation. The geometry of the model nanopore is based on the SEM image of the nanopore which are $d = 100$ nm, $\theta = 5^\circ$, $l_{\text{nanopore}} = 25$ μm (Figure 1a and S1-2). The surface charge density of the nanopore was set as -10 mC/m². To accelerate the calculation, the yellow line of boundary ⑤ is defined as surface of glass

with surface charge density of -10 mC/m^2 while green line of boundary ④ set as surface of glass without surface charge density. The 2D axisymmetric geometry of a nanopore for the simulation is shown in Figure S5. Note that these geometries are not drawn to scale.

Figure S5. The 2D axisymmetric geometry, the mesh for the finite-element simulation, and the boundary settings of the nanopore for the simulation of the potential distribution. Blue lines (①, ②, ③, ⑥, ⑦, ⑧) are the internal bulk solution and external bulk solution. Green line of ④ is the surface of glass. Yellow line of ⑤ is the surface charge of glass.

In order to simplify the simulation, here, both the outside and inside solution are set as 10 mM KCl. The dynamics of the ions in the channel is governed by the Poisson-Nernst-Planck (PNP) equations which relate surface charge with ionic fluxes and corresponding conductivity distribution within a glass nanopore. The ionic flow is computed by the Nernst-Planck (NP) equation (eq. 1) where the diffusion, migration, and convection terms are included.

$$\mathbf{J}_i = -D_i \nabla c_i - \frac{Fz_i}{RT} D_i c_i \nabla \Phi + c_i \mathbf{u} \quad (1)$$

In equation (eq. 1), \mathbf{J}_i is the ionic flow vector, F is the Faraday's constant, T is the absolute temperature, Φ is the potential, \mathbf{u} is the position-dependent fluid velocity, respectively. D_i , C_i , and Z_i represent the diffusion coefficient, the concentration, and the charge of species i in solution, respectively.

As neither species are transported into or out of the surface of glass (e.g. boundaries ②, ③, ④, ⑤, ⑥ and ⑦ in 2D axial symmetric model, Figure S5), there is no normal flux; the boundary condition is set as:

$$\mathbf{N} \cdot \mathbf{J}_i = 0$$

The relationship between the electric potential and ion concentration is described by Poisson equation, eq. 2.

$$\nabla^2 \Phi = -\frac{F}{\epsilon} \sum_i z_i c_i \quad (2)$$

Where ϵ is the dielectric constant of the medium.

The flow distribution is given by the Navier-Stokes equation, eq. 3, describing the pressure and electrical force driven flow.

$$\rho \nabla \mathbf{u} = \frac{1}{\rho} (-\nabla p + \eta \nabla^2 \mathbf{u} - F (\sum_i \sigma_i c_i) \nabla \Phi) \quad (3)$$

Here, ρ and η are the density and viscosity of the fluid, and p is the pressure.

ρ sets as $1 \times 10^{-3} \text{ kg/m}^3$ and η sets as $1 \times 10^{-3} \text{ Pa} \cdot \text{s}$.

Numerical solution of the equations involves discretization, which uses a mesh. Representative meshes for the finite-element simulations are shown in Figure S5. In each model, right part shows the entirety of the domain and right part illustrates the region at the tip of nanopore. The finite element simulations were carried out with COMSOL Multiphysics 5.2a (COMSOL Inc., Burlington, MA, USA) operated on a Lenovo P500 workstation (Intel(R) Xeon(R) CPU E5-1620 v3@3.50GHz, 4core, 32GB RAM)

Therefore, the simulation results in Figure S6 shows the electrical potential distribution

along the centerline axis of a 100 nm diameter nanopore with a surface charge -10 mC/m^2 for applied biases potential at -1 V and 1 V . The FEM demonstrates that the largest fraction of the bias potential drops at the narrowest tip of nanopore with approximately region of $15 \text{ }\mu\text{m}$ in length.

Figure S6. Electric potential along the center-line z axis ($r = 0$) of a 100 nm diameter nanopore in 10 mM KCl as a function of applied bias (ΔV) across the nanopore at -1 V and $+1 \text{ V}$, respectively.

In this revision, the simulation results are supplied in Main Text as follows:

“The finite element method simulation (FEM)^{19,31} was used to demonstrate that the largest fraction of the bias potential drops at the narrowest tip of nanopore with approximately region of $15 \text{ }\mu\text{m}$ in length (Figure S5-S6), which induced a strong electric field at the tip confinement for generating the electrochemical surface tension effects. The detailed description of the FEM simulation is shown in Supplementary Information.”
(Page 6, Line 3-8, Main Text)

The description of the FEM simulation has been supplied in Page 7-9, Supplementary Information.

Q5: As shown in Figure 2, the author has proved that the AIEgens could be manipulated inside nanopipette. However, what about the reversible usage of a nanopipette? How long does the nanopipette persevere in an experiment? These two points are very important for demonstrating the applicability of this method.

A5: Thank you very much for your comments. At present stage, one quartz nanopore could endure a 30-minute long performance for electrochemical manipulation of the DMTPS-DCV's emission due to the inevitable volatilization of the acetonitrile, which challenges its repeatable usage of a single nanopore. In our group, we endeavor to enhance long time performance and reusability of this method in our following studies.

We have added the discussions and related results in **Main Text** and **Supplementary Information** as follows:

“The quartz nanopore remained good capability of the continuous reversible modulation for around 30 min. In our following studies, we endeavor to largely eliminate the volatilization of the acetonitrile for enhancing long time performance and reusability of this method.” (Page 9, Line 15-18, Main Text)

Q6: The author stated that “Previous studies suggest that the application of voltage across the liquid/liquid interface changes the surface tension, previous studies suggest that the application of voltage across the liquid/liquid interface changes the surface tension.... Therefore, we applied bias potential to trigger the motion of DMTPS-DCV solution in/out of the nanopore.” They attribute the electrical motion of the AIEgen to the liquid/liquid interface changes. Is it possible that the electro-osmotic flow or electrophoretic forces affects the motion of the AIEgen?

Q7: Since the AIEgen visualizes the fluidic motion from acetonitrile solution to the aqueous solution. Is it possible for this technique to explore the more fluid mechanics inside nanopipettes? The author was encouraged to provide more discussions about

the fluid flow inside the nanopipette.

A6&A7: The physicochemical processes involved in the electrochemical modulation of AIEgens are based on the electrowetting. The liquid flow is largely determined by the surface tension effects. As describe in previous studies (*J. Phys. Chem.*, 1996, 100 17881–17888; *Phys. Chem. Chem. Phys.*, 2010, 12, 1685-1697; *Phys. Chem. Chem. Phys.*, 2010, 12, 1685-1697; *Electrochim. Acta* **2000**, 45, 2647-2662), the relatively high concentration of the electrolyte both in organic and aqueous solution could across the liquid/liquid interface. For an optimal syringe performance, the ion-transfer current should be minimized by using the hydrophobic organic electrolyte. On the other hand, the electrokinetic including the electrophoresis and electroosmosis occur within the confinement of the nanopore. If applying this method to the charged AIEgens nanopipette, one should ensure that the changes of the surface tension which drives the the ingress of organic solution or the expulsion of the aqueous solution should faster that the electrokinetic processes. In this experiment, the uncharged molecule of DMTOS-DCV was manipulated inside the nanopore, which is largely determined by the surface tension across the liquid/liquid interface rather than the electrokinetic processes. If employed the charged AIEgen, this method could be used to explore the electrokinetic processes involved fluid mechanics inside confined nanopores. In this revision, we have added the discussion about the electrokinetic processes for modulating the charged the AIEgens as follows:

“If extending this method to the charged AIEgens, one should ensure that the changes of the surface tension which drives the the ingress of organic AIE solution or the expulsion of the aqueous solution should faster that the electrokinetic processes²⁷. Additionally, the electrostatic adsorption of the positively charged AIEgens into the negatively-charged inner wall of nanopore may prevent one from the electrochemically modulation of the emissions. This limitation could be overcome by functionalizing the inner surface of nanopore, e.g. salinization to form a hydrophobic surface²⁷.” (Page 12, Line 2-8, Main Text)

“Especially, the real-time visualization of the AIE emission in a nanopore could help

us to explore the fluid mechanics inside confinement including the local pressure effects and electrokinetic effects.” (Page 12, Line 11-13, Main Text)

Q8: *The author suggests that “the quartz nanopore provides a suitable platform for the AIE which would be used for on-site environmental monitoring and diseases diagnosis such as fluorescence image-guided surgery.” Could the author provide more charming pictures for applying this strategy in AIEgens or detailed example for the applications? These discussions would help the readers in different fields to quickly get the inspiration from their novel strategy.*

A8: Under the applied bias potential, the motion of AIEgens could be triggered to move in/out of the nanopore. Therefore, this method could control the organic soluble AIEgens to electrochemically behavior “on-to-off” and “off-to-on” emission. By using this method, the emission of organic soluble AIEgens could be manipulated cell imaging, bacteria imaging, photodynamic therapy, protein/DNA detection and etc.. The possible molecule could include but not limited to derivatives of tetraphenylethene, hexaphenylsilole, diphenylethylene anthracene and diphenyl fluorene. In order to improve the compatible of our method to various fluorescent spectroscopy, in our lab, we are continuing develop the portable electrochemical instrument. Therefore, the commercial device based on nanopore-AIE method could be expected which will be applied in wide AIE applications.

In this revision, we have added the following statements to describe the prospect of this method.

“By further incorporating with the efficient AIEgen, our method would facilitate the delivery of femtoliter AIEgen into the single cell for the anticipated applications of high resolution single cell imaging, single cell theranostic, single cell diagnostics, bacteria imaging, photodynamic therapy, protein/DNA detection and etc..” (Page 11, Line 23-25 & Page 12, Line 1-2, Main Text)

“In order to improve the compatible of our method to various fluorescent spectroscopy, in our lab, we are continuing develop the specialist portable

electrochemical instrument for this technique.” (Page 12, Line 14-15, Main Text)

Q9: *Since the nanopore confinement provides the new way to manipulate AIEgens. The author should give more details for the experimental chamber design, the optical configuration as well as the the position of electrodes.*

A9: Thank you very much for your suggestions. In this revision, we have supplied the detail design of the experimental chamber, the optical configurations and the position of electrodes as follows:

“The quartz nanopore is placed on the designed cover glass (see details in Figure S1).”
(Page 4, Line 15-16, Main Text)

Figure S1. The designed cover glass with a outside chamber for filling with the DMTPS-DCV acetonitrile solution. The outside chamber is sealed with the acrylic ab adhesive. (Page 3, Supplementary Information)

Figure S2. Schematic configuration for integrating the inverted fluorescence microscopy with the ultralow ionic current acquisition system. The synchronized collection of electrical and optical signals was guaranteed by the multichannel design of the analog to digital converter (ADC), then monitored by the computer. (Page 4, Supplementary Information)

Q10: As describe in Supplementary Information, “The potential is applied using a pair of Ag/AgCl electrodes. The electrical measurements were performed on a patch clamp amplifier” However, Figure 1a and 3a lacks symbol of electric supply, amperemeter and the ground definition. According to the presented figures, it is confused that if the two electrodes link to one amplifier or two dependent amplifiers. The author should link two electrodes together to one amplifier.

A10: Thank you very much for your suggestions. The two Ag/AgCl electrodes are connected to the same amplifier. In this revision, we have added the symbols of electric supply, and the ground definition in Figure 1a and Figure 3a. Moreover, the

details for the connection between two electrodes and the amplifier is supplied in Figure S2. The revised figures are shown as follows:

Figure 1 | Visualizing the dynamic emission of DMTPS-DCV regulated by the electrochemically confinement of a quartz nanopore. (a) The instrumentation for the visualization of DMTPS-DCV emission inside a quartz nanopore. The quartz nanopore is placed on the designed cover glass (see details in Figure S1). The inside and outside nanopore filled with aqueous solution and 10 μ M DMTPS-DCV acetonitrile solution, respectively. The 10 mM TBAPF₆ and 10 mM KCl were used as organic and aqueous electrolyte to conduct the ionic flow. Two Ag/AgCl electrodes were used to apply the potential while the acetonitrile side is defined as virtual ground. The ionic current is recorded by an ultrasensitive electrochemical amplifier (see supplementary information for details). The light of $\lambda = 450-490$ nm was used to excite the DMTPS-DCV. All the fluorescent images were recorded by EMCCD at exposure time of 50 ms and gain of 300. Detailed Schematic configuration for synchronized collection of electrical and optical signals are shown in Figure S2. Insert: the SEM image of the quartz nanopore with a diameter of approximately 100 nm. (b) The electrochemical manipulation of DMTPS-DCV at applied bias potential of 1 V. Left: the DMTPS-DCV solution moves from outside nanopore to the inside nanopore at 1V, leading to an intensive emission. Right: the fluorescent images of the “on-to-off”

transition. (c) The electrochemical manipulation of DMTPS-DCV at applied bias potential of -1 V. Left: the DMTPS-DCV solution moves from inside nanopore to the outside nanopore at -1V, leading to a decrease in emission. Right: the fluorescent images of the “off-to-on” transition.

motion of DMTPS-DCV inside a confined nanopore, which are recorded within 5 s. (c) The electrochemical manipulation of DMTPS-DCV at applied bias potential of -1 V. Left: the applied potential of -1 V drove the movement of DMTPS-DCV aggregates from the wider lumen to the confined tip. Right: Fluorescent images recorded at -1 V, illustrating the “off-to-on” motion of DMTPS-DCV. The bright pixel moves from upper to tip within 35 s. The length of the quartz nanopore shown in each fluorescent image is 65 μm which could be divided into three confinements of I ($z = 0 \sim 16 \mu\text{m}$), II ($z = 16 \sim 56 \mu\text{m}$) and III ($z = 56 \sim 100 \mu\text{m}$), respectively. The white arrow indicates the strongest location of DMTPS-DCV emission.

Figure 3 | Cell injection of DMTPS-DCV using electrochemically nanopore manipulation. (a) The illustration for the cell injection. The DMTPS-DCV is sucked into the nanopore at 1 V, then the nanopore is translated toward the cell and penetrates through the cell membrane. The DMTPS-DCV is injected inside the cell at -1 V. (b) The fluorescence micrographs (upper) and dark-field images (bottom) of immobilized MCF-7 cell. Left: After quartz nanopore punches into the cell, the DMTPS-DCV is hold back away from the nanopore tip at 1 V, leading to the less emission at the nanopore tip; Right: DMTPS-DCV is electrochemically driven into the single cell at -1V, which enlightens the intra cell. The concentration of DMTPS-DCV in this injection is 10 mM.

Reviewers' comments:

Reviewer #1 (Remarks to the Author):

I am satisfied with the detailed answers for my comments. I think authors made careful revisions. I recommend publication of this paper essentially as it stands.

Reviewer #2 (Remarks to the Author):

There is no doubt that AIE is an exciting phenomenon and AIE molecules have numerous potential applications. However, my major reservation about this work is the application of the AIE nano-device in biological imaging. The authors tried to address my concerns, but did inaccurately and incorrectly. The following statements in the rebuttal letter are false:

1. “These optical controls require high laser load to achieve high enough off/on ratio to enable single-molecule localization for super-resolution microscopy thus resulting in unexpected background, more phototoxicity and complicated operations ...”
2. “...most of these chromophores demonstrate poor cell permeability and require external additives... these molecules and nanoparticles are hardly to be directly used in the single-cell imaging...”
3. “the cell cultivation using luminescent agents with high concentration was an essential step for the cell imaging which may put a higher risk of cell toxicity.”
4. "The small diameter of our nanopore (~100 nm) causes less damage to the single cell"

Photoactivatable fluorescent probes have been widely used for superresolution imaging based on the switching and localization of single molecules. Specifically, photoactivatable fluorescent proteins (PAFPs) are genetically encoded. PAFPs can target specific biological structures and molecules in cell ("Photoactivatable fluorescent proteins for diffraction-limited and super-resolution imaging", *Trends in Cell Biology*, V.19, 2009, 555-565; "Fluorescent proteins for live-cell imaging with super-resolution", *Chem. Soc. Rev.*, 2014, 43, 1088-1106; "A Photoactivatable Probe for Super-Resolution Imaging of Enzymatic Activity in Live Cells", *J. Am. Chem. Soc.* 139, 37, 13200-13207).

The large size (~100 nm) of the nanopores reported in this manuscript will cause serious problem in biological imaging, particularly in vivo imaging of living cells because their large size could alter the mobility of the molecule of interest and the functionality of targeted structures. This is

already a significant concern in the application of fluorescent quantum dots (typically ~15–20 nm in diameter) for in vivo sensing applications and FRET-based biosensing ("Biocompatible Quantum Dots for Biological Applications", <https://doi.org/10.1016/j.chembiol.2010.11.013>; "Limitations of Qdot labelling compared to directly-conjugated probes for single particle tracking of B cell receptor mobility", *Scientific Reports*, V.7, Article number: 11379 (2017)). The authors must address the issue how to shrink the size of nanopore to avoid altering the mobility of the molecule of interest and the functionality of targeted structures in living cells.

Reviewer #3 (Remarks to the Author):

I have carefully read the point-by-point response to referees. All my concerns have been addressed. I recommend publication in its current form.

Responses to Reviewer #2

Thank you very much for your comments and your concerns about the potential applications of our methods in cell imaging. Indeed, the super-resolution cell imaging is one of the goal in AIE field since recent attempts just achieved the super-resolution fluorescence imaging of cancer cells based on stimulated emission depletion (STED) nanoscopy and AIE nanoparticles (*Adv. Mater.*, 2017, 29, 1703643). However, **our manuscript aims to manipulate the emission of AIEgens reversibly and dynamically** since it is still a big challenge in reversible manipulating the emission of AIEgens from the initial dispersed state to the ultimate aggregated state. We further applied this dynamic manipulation to achieve a target delivery of AIEgens into a single cell. Therefore, **we are afraid that the “super-resolution cell imaging” is beyond the scope of our manuscript.** In this response, we would like to address your concerns and clarify the misunderstanding from following aspects.

1. The quartz nanopore is a **controllable delivery tool** of AIEgens rather than a probe for targeting the structures.

Concerns from the Reviewer #2: “The large size (~100 nm) of the nanopores reported in this manuscript will cause serious problem in biological imaging, particularly in vivo imaging of living cells because their large size could alter the mobility of the molecule of interest and the functionality of targeted structures.”

Response: After controllable delivering of AIEgens into the single cell, the nanopore is recessed back into the solution rather than continuous remain side the single cell. The whole injection process inside the single cell completes within 1 min. **The nanopore is not responsible for the target specific structure or molecules of interests inside the single cell.** The mobility of the molecule of interests or/and the functional structure is more likely and largely influenced by its specific probing chromophore rather than the delivering methods. Our results show that there is no change in the cell shape, size, or position after the nanopore injection, which demonstrates the excellent cell viability of our methods (Figure S14). Moreover, our previous study demonstrates that the single cell could tolerant to the three consecutive

punches of the ~ 90 nm nanopore (*J. Am. Chem. Soc.*, 2018, 140, 5385). In addition, a previous study demonstrates the normal behavior of the cell and a healthy cell cycle after 1-min 100 nm nanopore punching at 1 V (*Sci. Rep.*, 2017, 7, 41277). Note that this conclusion results from the statistical analysis of cell viability for 24 h after the nanopore injection. **Therefore, the ~ 100 nm nanopore has a significantly low impact on the functionality of the injected cells.** To clearly describe the injection process of our methods, we have added the following statements in Main Text:

“Afterwards, the quartz nanopore is recessed back into the solution rather than continuously remain side the single cell. The whole injection process inside the single cell completes within 1 min.” (Page 10, Line 19-20, Main Text)

“Moreover, a previous study demonstrates the normal behavior of the cell and the healthy cell cycle after 1-min 100 nm nanopore punching at 1 V.³³ Therefore, the ~ 100 nm nanopore has a significantly lower impact on the functionality of the injected cells.”

We have added a new reference as Reference 33 to demonstrate the less cell damage induced by our methods.

“33.Simonis, M., Hübner, W., Wilking A., Huser, T. & Henning S. Survival rate of eukaryotic cells following electrophoretic nanoinjection, *Sci. Rep.* **7**, 41277 (2017).”

2. The quartz nanopore does not bind with the target inside the single cell

*Concerns from the Reviewer #2: “This is already a significant concern in the application of fluorescent quantum dots (typically ~15–20 nm in diameter) for in vivo sensing applications and FRET-based biosensing (“Biocompatible Quantum Dots for Biological Applications”, <https://doi.org/10.1016/j.chembiol.2010.11.013>; “Limitations of Qdot labelling compared to directly-conjugated probes for single particle tracking of B cell receptor mobility”, *Scientific Reports*, V.7, Article number: 11379 (2017)).”*

Responses: We have carefully read the two references suggested by Reviewer 2. *Sci. Rep.*, 2017, 7, 11379 adopted the streptavidin attached quantum dots which was then conjugated to the monovalent antigen-binding fragment (Fab) of antibodies. This

quantum dots directly bounded with the B cell receptors due to the target recognition of Fab. The results show that the bounded quantum dots will hinder the lateral mobility of the receptor while small organic fluorophore Cy3 hardly induce the difference in receptor mobility. In our manuscript, the quartz nanopore did not bind to any of the target receptor. It controllably injected the small organic AIE fluorophore into the cell. As a proof of concept experiment, we used the 1,1-dimethyl-2,5-bis(4-benzylidenemalononitrile)-3,4-diphenylsilole (DMTPS-DCV) as an AIE model molecule for single cell injection and imaging. Therefore, the fluorophore sensing molecule used here is a small organic AIE molecule, which may not induce the serious difference of lateral mobility for the cell component according to the conclusion from the Reviewer #2 suggested reference “*Scientific Reports, 2017, 7, 11379*”. In all accounts, the impact of nanomaterials and/or organic fluorophore on the intrinsic dynamics of the biomolecules inside single cell are not the topic of this manuscript.

Another paper the Reviewer 2 suggested is *Chem Biol. 2011, 18, 10–24*. This is a review paper summarized the development of quantum dots for biological applications, which suggests that “for single protein tracking experiments in which one is trying to elucidate the velocities of the protein, one would not like the mass of the quantum dot label to alter the intrinsic dynamics.” In our experiments, we do not use quantum dot for the single cell imaging. Moreover, the delivered AIEgen by our methods which does not further applied in single protein tracking experiments. We only show the possibility of our methods for electrochemically controllable inject the AIEgen into the single cell for the imaging. **The topic of how to maintain the intrinsic dynamics of the biomolecules in the single cell imaging is beyond the scope of our manuscript.** We felt that it is inappropriate for us to discuss such a huge topic in this manuscript. Hope you could understand.

3. The quartz nanopore does not aim to achieve the super-resolution imaging

Concerns from the Reviewer #2: Photoactivatable fluorescent probes have been widely used for superresolution imaging based on the switching and localization of

single molecules. Specifically, photoactivatable fluorescent proteins (PAFPs) are genetically encoded. PAFPs can target specific biological structures and molecules in cell ("Photoactivatable fluorescent proteins for diffraction-limited and super-resolution imaging", Trends in Cell Biology, V.19, 2009, 555-565; "Fluorescent proteins for live-cell imaging with super-resolution", Chem. Soc. Rev., 2014, 43, 1088-1106; "A Photoactivatable Probe for Super-Resolution Imaging of Enzymatic Activity in Live Cells", J. Am. Chem. Soc. 139, 37, 13200-13207).

Response: Thank you very much for providing us these very good papers about fluorescent proteins. Indeed, the fluorescent proteins show excellent ability in super-resolution imaging. However, this manuscript does not aim to achieve the super-resolution imaging. As presented in our abstract, this manuscript addresses the challenge of “However, all the AIE fluorogens (AIEgens) hardly recover into the initial dispersed state after illuminating at the ultimate aggregated state, which limits AIEgenes to achieve reversible sensing and reproducible devices.” (Abstract, Page 1, Line 4-6, Main Text). In our manuscript, we take the advantage of the confined space in the quartz nanopore to achieve a nanopore-size-dependent restriction of AIEgens for the reversible conversions of “on-to-off” and “off-to-on” emission. We further applied this dynamic manipulation for a target delivery of the AIEgen into a single cell. As discussed in the “discussions” part, “Since the AIE emission could be electrochemically controlled, it allows for a femtoliter cell injection of the AIEgen with poor cell permeability which does not need more sophisticated modifications.” (Line 1-2, Page 17, Main Text). Therefore, our methods provide the efficient way for delivery the AIEgen into the single cell instead of aiming for the “super-resolution cell imaging”. To avoid overselling of this method, all the Results part did not contain any description of “Super-resolution cell imaging”. We also emphasized that “By further incorporating with the efficient AIEgen, our method would facilitate the delivery of femtoliter AIEgen into the single cell for the anticipated applications of high resolution single-cell imaging...” (Line 2-4, Page 17, Main Text).

All the fluorescent nanomaterials, organic molecule dye and fluorescent proteins have achieved great development in super-resolution imaging during these years (see

Nature, 2017, 543, 229; *Nat. Chem.*, 2013, 5, 132; *Adv. Mater.*, 2017, 29, 1703643).
However, our manuscript focuses on providing organic AIE molecules a controllable manipulation for its potential applications.

4. The responses for the Reviewer #2-selected statements in our 1st round rebuttal letter

Selected Statement 1 “These optical controls require high laser load to achieve high enough off/on ratio to enable single-molecule localization for super-resolution microscopy thus resulting in unexpected background, more phototoxicity and complicated operations ...”

Responses: We’re sorry for this unclear statement at our last Rebuttal Letter. This statement is related to the single-molecule localization microscopy which is one of the super-resolution microscopes. The whole statements in our 1st round rebuttal letter is as follows:

“In super-resolution microscopy as single-molecule localization microscopy (SMLM), those photoactivatable molecules and the quantum dots usually switched between their bright and dark states. These optical controls require high laser load to achieve high enough off/on ratio to enable single-molecule localization for super-resolution microscopy thus resulting in unexpected background, more phototoxicity and complicated operations (*Photochem. Photobiol. Sci.* 2011, 10, 499).” (A3, Page 21, 1st round rebuttal letter)

Numerous studies show that the super-resolution microscopy such as SMLM, STED, STORM) enquires high laser load comparing to the traditional confocal microscopy or Total internal reflection fluorescence (TIRF) microscopy, see *Nature Communications* 2017, 8, 2090; *Biophys. J.* 2014, 107, 1777; *Biochemistry*, 2017, 56, 5194; *Chem. Rev.*, 2017, 117, 7478. Moreover, a review of “Photoactivatable fluorescent proteins for diffraction-limited and super-resolution imaging” (*Trends Cell Biol.*, 2009, 19, 555) suggested by the Reviewer #2 also summarized the applications of photoactivatable fluorescent proteins on the photoactivated localization microscopy (PALM) and stochastic optical reconstruction microscopy (STORM) techniques for

super-resolution imaging. Both two techniques require the higher laser load (Science, 2006, 313, 1642; Nat. Methods, 2006, 3, 793) of about tens mW to W compared to the confocal microscopy ($\sim < 0.5$ mW). Besides, the other papers suggested by the Reviewer #2 of “A Photoactivatable Probe for Super-Resolution Imaging of Enzymatic Activity in Live Cells” (*J. Am. Chem. Soc.*, **2017**, 139, 13200) and “Fluorescent proteins for live-cell imaging with super-resolution” (*Chem. Soc. Rev.*, 2014, 43, 1088) also demonstrate the designed fluorescent proteins for super-resolution imaging by STED, PALM and STORM. Especially, *Chem. Soc. Rev.*, 2014, 43, 1088 summarized that “the activation and localization steps are repeated, until the population of single activated molecules becomes negligible even at high laser powers (10^2 – 10^4 times).” Therefore, our previous responses are proper and objective for the super-resolution microscopy. More important, **we did not involve any discussions about the super-resolution microscopy in our Main Text and Supplementary Information since this topic is beyond the scope of this manuscript.**

Selected Statement 2 “...most of these chromophores demonstrate poor cell permeability and require external additives... these molecules and nanoparticles are hardly to be directly used in the single-cell imaging...”

Responses: We’re sorry for the misunderstanding in our 1st round rebuttal letter. This statement describes the limitation of organic small chromophores and nanoparticles. The previous studies show that the external additives could facilitate these chromophores to achieve a good cell permeability (*Nat. Methods* 2011, 8, 1027; *Nat. Chem.* 2014, 6, 681). In our Main Text, we describe the advantage of our methods as “Since the AIE emission could be electrochemically controlled, it allows for a femtoliter cell injection of the AIEgen with poor cell permeability which does not need more sophisticated modifications.” (Page 12, Line 1-2). This statement is within the scope of our presented method in this manuscript.

Selected Statement 3 “the cell cultivation using luminescent agents with high concentration was an essential step for the cell imaging which may put a higher risk

of cell toxicity.”

Responses: We're sorry for the misunderstanding at our 1st round rebuttal letter. This sentence describes the current understanding for the small organic fluorophore including AIEgen. As shown in previous researches, the small organic fluorophore has the potential cell toxicity at high concentration (*Mol. Imaging, 2009, 8, 7290*). Nevertheless, this statement did not include in the Main Text and Supplementary Information of our manuscript. In our Main Text, we have clearly present the anticipated applications of our methods as follows:

“Since the AIE emission could be electrochemically controlled, it allows for a femtoliter cell injection of the AIEgen with poor cell permeability, which does not need further sophisticated modification. By further incorporating with the efficient AIEgen, our method would facilitate the delivery of femtoliter AIEgen into the single cell for the anticipated applications of high resolution single-cell imaging, single-cell diagnostics, single-cell theranostics, bacteria imaging, photodynamic therapy, protein/DNA detection and etc..” (*Page 11, Line 22-25 & Page 12, Line 1-2, Main Text*)

Selected Statement 4 “The small diameter of our nanopore (~100 nm) causes less damage to the single cell”

Responses: As we described in our first aspect in this Rebuttal letter, a previous study demonstrates the normal behavior of the cell and a healthy cell cycle after 1-min 100 nm nanopore punching at 1 V (*Sci. Rep. 2017, 7, 41277*). The whole nanopore injection process from the cell punching, AIEgen injection and nanopore retraction was completed within 1 min. Moreover, the fluorescence micrograph of the immobilized MCF-7 cell before and after injection show that the shape, size, or position of the cell did not changed after the nanopore injection (Figure S14). Therefore, our presented ~ 100 nm nanopore induces less damage to the single cell.

Based on above responses, we feel that the statement in our Main Text and Supplementary Information is appropriate and logic. The conclusion reached by our

results is within the scope of this manuscript.

Reviewer #4 (Remarks to the Author):

Long and coworkers reported a novel method for real-time manipulating and visualizing the dynamic aggregation-induced emission process within a confined quartz nanopore. The nanopore could be gradually illuminated from the constricted tip to the inside confinement in a highly reversible and reproducible manner. This method could not only achieve the reversible manipulation of the AIE emission in “on-to-off” and “off-to-on” manner but also could realize the electrochemical loading of femtoliter AIEgen solution. In addition, this method allows for a femtoliter cell injection of the AIEgen with poor cell permeability and does not need more sophisticated modifications, which could be used for the on-site environmental monitoring and diseases diagnosis such as fluorescence image-guided surgery. The results of this work is interesting and important. I recommend the publication of this work with minor revisions.

1. The authors claimed that ‘Since the cylindrically shaped cavity of the nanopore owns a larger volume than its conical tip, the DMTPS-DCV aggregates become more dispersed to exhibit weaker emission’, which suggested that the weaker emission is due to the low concentration of the DMTPS-DCV aggregates. Is the “on-to-off” phenomenon really the visualization of the dynamic aggregation-induced emission process (from aggregates to molecular species or from molecular species to aggregates), or just the change of aggregates concentration? please add some comments to clarify this concern.

2. Water is supposed to be a poor solvent of DMTPS-DCV, and DMTPS should form aggregates in water solution. How did the authors conduct the control experiment when both the outside and inside of nanopore were filled with the aqueous solution?

3. In the caption of Figure 1c, the authors stated that “The length of the quartz nanopore shown in each fluorescent image is 65 μm which could be divided into three confinements of I ($z = 0\sim 16\ \mu\text{m}$), II ($z = 16\sim 56\ \mu\text{m}$) and III ($z = 56\sim 100\ \mu\text{m}$), respectively.” It is strange that the total length is 65 μm and it contains a section of III which range from 56-100 μm . Please double check this technical issue.

4. In the abstract, “AIE fluorogens (AIEgens)” should be changed to “AIE luminogens (AIEgens)”, since the emission of AIEgens including not only fluorescence.

5. Also in the abstract, some incorrect spelling such as “AIEgenes” and “AIEgene”, which should be “AIEgens” and “AIEgen”, respectively.

Responses to Reviewer #4

Thank you very much for your constructive comments and suggestions. Our manuscript has been improved according to your guidance. In this revision, we have provided more clearly statement about the dynamic motion of *DMTPS-DCV* inside nanopipette and corrected the typos thoroughly through the manuscript. We use the tracked changes feature of Microsoft Word to make all these changes. The details of the revision are listed as follows:

***Q1:** The authors claimed that ‘Since the cylindrically shaped cavity of the nanopore owns a larger volume than its conical tip, the *DMTPS-DCV* aggregates become more dispersed to exhibit weaker emission’, which suggested that the weaker emission is due to the low concentration of the *DMTPS-DCV* aggregates. Is the “on-to-off” phenomenon really the visualization of the dynamic aggregation-induced emission process (from aggregates to molecular species or from molecular species to aggregates), or just the change of aggregates concentration? please add some comments to clarify this concern.*

***A1:** Thank you very much for your concerns. As setting the optimal bias potential to 1 V, the acetonitrile solution of *DMTPS-DCV* is driven into the inside aqueous solution from the exterior. Therefore, the quartz nanopore fills with 26 fL of 10 μ M *DMTPS-DCV* acetonitrile solution (position I) followed by the aqueous solution (position II and III) from the tip side to the upper end. At this stage, the dissolved *DMTPS-DCV* does not exhibit obvious emission at position I. Then, with the continuous applied potential of 1 V, the *DMTPS-DCV* is driven into the inside poor solvent of acetonitrile. Due to the confined effects of quartz nanopore, the solubility gradient makes *DMTPS-DCV* gradually aggregate from molecule species. As a consequence, there is a strong emission occurring at the position II of quartz nanopore. As the time elapses, the brightest pixel in the individual frame of fluorescence image shifts to the upper end of the nanopore (moving in Position II in Figure 1b). Afterwards, the *DMTPS-DCV* aggregates become more dispersed due to the*

decreased concentration in the large volume of Position III, which exhibit weaker emission. Therefore, the “on-to-off” phenomenon includes the dynamic aggregation-induced emission process followed by the concentrated-dependent emission of the aggregation state.

In this revision, we have revised the corresponding sentence to clearly present our results as follows:

“Consequently, the brightest pixel in the individual frame of fluorescence image shifts to the upper end of the nanopore (Position II in Figure 1b), which is due to the dynamic aggregation-induced emission process. Since the cylindrically shaped cavity of the nanopore owns a larger volume than its conical tip, the DMTPS-DCV aggregates become more dispersed to exhibit weaker emission at Position III.”

Q2: Water is supposed to be a poor solvent of DMTPS-DCV, and DMTPS should form aggregates in water solution. How did the authors conduct the control experiment when both the outside and inside of nanopore were filled with the aqueous solution?

A2: Thank you very much for your concerns. In our control experiments, the 10 μM DMTPS-DCV aqueous solution was added in the outside nanopore. As water is a poor solvent of DMTPS-DCV, the outside solution of nanopore contains the aggregates of DMTPS-DCV. As shown in Supplementary Figure 11, the bright spot in the outside solution indicates the aggregates of DMTPS-DCV in aqueous solution. However, the inside solution does not show any bright spots. This experiment supports that the aggregates of DMTPS-DCV could not be egressed into the quartz nanopipette and further exhibit the fluorescence. Therefore, only the molecule species of DMTPS-DCV in good solvent of acetonitrile could be the dynamic restricted along with the nanopore, which promotes the emission of DMTPS-DCV.

Q3: In the caption of Figure 1c, the authors stated that “The length of the quartz nanopore shown in each fluorescent image is 65 μm which could be divided into three confinements of I ($z = 0\sim 16 \mu\text{m}$), II ($z = 16\sim 56 \mu\text{m}$) and III ($z = 56\sim 100 \mu\text{m}$), respectively.” It is strange that the total length is 65 μm and it contains a section of

III which range from 56-100 μm . Please double check this technical issue.

A3: Thank you very much for your corrections. We have corrected this mistake and revised the caption of Figure 1 as follow:

“The length of the quartz nanopore shown in each fluorescent image is 74 μm which could be divided into three confinements of I ($z = 0 \sim 16 \mu\text{m}$), II ($z = 16 \sim 56 \mu\text{m}$) and III ($z > 56 \mu\text{m}$), respectively.”

Q4: *In the abstract, “AIE fluorogens (AIEgens)” should be changed to “AIE luminogens (AIEgens)”, since the emission of AIEgens including not only fluorescence.*

A4: Thank you very much for your comments. The related sentence has been corrected as “However, all the AIE luminogens (AIEgens) hardly recover into the initial dispersed state after illuminating at the ultimate aggregated state,”. Also, the first sentence in the introduction has been revised as “The emerged aggregation-induced emission luminogens (AIEgens) show the characteristics of high emission efficiency in the aggregated state with advantages of high photostability and large Stokes’ shift.^{1,2}”

Q5: *Also in the abstract, some incorrect spelling such as “AIEgenes” and “AIEgene”, which should be “AIEgens” and “AIEgen”, respectively.*

A5: Thank you very much for your corrections. All the words of “AIEgenes” and “AIEgene” have been thoroughly corrected as “AIEgens” and “AIEgen” through the manuscript.